# FedPAGE: A Fast Local Stochastic Gradient Method for Communication-Efficient Federated Learning

## Abstract

Federated Averaging (FedAvg, also known as Local-SGD) (McMahan et al., 2017) is a classical federated learning algorithm in which clients run multiple local SGD steps before communicating their update to an orchestrating server. We propose a new federated learning algorithm, FedPAGE, able to further reduce the communication complexity by utilizing the recent optimal PAGE method (Li et al., 2021). We show that FedPAGE uses much fewer communication rounds than previous local methods for both federated convex and nonconvex optimization. Concretely, 1) in the convex setting, the number of communication rounds of FedPAGE is $O(\frac{N^{3/4}}{S\epsilon})$, improving the best-known result $O(\frac{N}{S\epsilon})$ of SCAFFOLD (Karimireddy et al., 2020) by a factor of $N^{1/4}$, where $N$ is the total number of clients (usually is very large in federated learning), $S$ is the sampled subset of clients in each communication round, and $\epsilon$ is the target error; 2) in the nonconvex setting, the number of communication rounds of FedPAGE is $O(\frac{\sqrt{N}+S}{S\epsilon^2})$, improving the best-known result $O(\frac{N^{2/3}}{S^{2/3}\epsilon^2})$ of SCAFFOLD (Karimireddy et al., 2020) by a factor of $N^{1/6}S^{1/3}$, if the sampled clients $S \leq \sqrt{N}$. Note that in both settings, the communication cost for each round is the same for both FedPAGE and SCAFFOLD. As a result, FedPAGE achieves new theoretical state-of-the-art results in terms of communication complexity for both federated convex and nonconvex optimization.

## 1 Introduction

With the rise in the proliferation of mobile and edge devices, and their ever-increasing ability to capture, store and process data, federated learning (Konečný et al., 2016b; McMahan et al., 2017; Kairouz et al., 2019) has recently emerged as a new machine paradigm for training machine learning models over a vast amount of geographically distributed and heterogeneous devices. Federated learning aims to augment the traditional centralized datacenter focused approach to training machine learning models (Dean et al., 2012; Iandola et al., 2016; Goyal et al., 2017) with a new decentralized modality that aims to be more energy-efficient, and mainly, more privacy-conscious with respect to the private data stored on these devices. In federated learning, the data is stored over a large number of clients, for example, phones, hospitals, or corporations (Konečný et al., 2016a;b; McMahan et al., 2017; Mohri et al., 2019). Orchestrated by a centralized trusted entity, these diverse data and compute resources come together to train a single global model to be deployed on all devices. This is done without the sensitive and private data ever leaving the devices.

One of the key challenges in federated learning comes from the fact that communication over a heterogeneous network is extremely slow, which leads to significant slowdowns in training time. While a centralized model may train in a matter of hours or days, a comparable federated learning model may require days or weeks for the same task. For this reason, it is imperative that in the design of federated learning algorithms one focuses special attention on the communication bottleneck, and designs communication-efficient learning protocols capable of producing a good model.

There are two popular lines of work for tackling this communication-efficient federated learning problem. The first makes use of general and also bespoke *lossy compression operators* to compress the communicated messages (such as local stochastic gradients) before they are sent over the net-

work (Mishchenko et al., 2019; Li and Richtárik, 2020; Li et al., 2020; Gorbunov et al., 2021; Li and Richtárik, 2021a), and the second line bets on increasing the local workload by performing *multiple local update steps*, e.g., multiple SGD steps, before communicating with the orchestrating server (Stich, 2020; Woodworth et al., 2020; Gorbunov et al., 2020; Karimireddy et al., 2020).

In this paper, we focus on the latter approach (multiple local update steps in each round) to alleviating the communication bottleneck in federated learning. One of the earliest and classical methods proposed in this context is FedAvg/local-SGD (McMahan et al., 2017; Sahu et al., 2018; Yu et al., 2019; Li et al., 2019; Haddadpour and Mahdavi, 2019; Stich, 2020; Gorbunov et al., 2020). However, the method has remained a heuristic until recently, even in its simplest form as local gradient descent, particularly in the important heterogeneous data regime (Khaled et al., 2020; Woodworth et al., 2020). Further improvements on vanilla Local-SGD have been proposed, leading to methods such as Local-SVRG (Gorbunov et al., 2020) and SCAFFOLD (Karimireddy et al., 2020). In particular, Gorbunov et al. (2020) also provide a unified framework for the analysis of many local methods in the strongly convex and convex settings.

## 1.1 OUR CONTRIBUTIONS

Although there are many works on local gradient-type methods, the communication complexity in existing works on local methods is still far from optimal. In this paper, we introduce a novel local method FedPAGE, significantly improving the best-known results for both federated convex and nonconvex settings (see Table 1). Now, we summarize our main contributions as follows:

1. We develop and analyze, FedPAGE, a fast local method for communication-efficient federated learning. FedPAGE can be loosely seen as a local/federated version of the PAGE algorithm of Li et al. (2021), which is a recently proposed optimal optimization method for solving smooth nonconvex problems. Our analysis of FedPAGE also recovers the optimal results of PAGE (see Theorem 1), thus FedPAGE substantially improves the best-known non-optimal result of SCAFFOLD (Karimireddy et al., 2020) by a factor of $N^{1/6}S^{1/3}$ (see Table 1 for more details).

2. For the convex setting, we provide the convergence results for FedPAGE in Theorem 2. Moreover, FedPAGE also improves the best-known result of SCAFFOLD (Karimireddy et al., 2020) by a large factor of $N^{1/4}$ (see Table 1).

3. Finally, we first conduct the numerical experiments for showing the effectiveness of multiple local update steps (see Section 6.1). The experiments indeed demonstrate that FedPAGE with multiple local steps $K \geq 1$ is better than that with $K = 1$ (no multiple local updates). Then we also conduct experiments for comparing the performance of different local methods such as FedAvg (McMahan et al., 2017), SCAFFOLD (Karimireddy et al., 2020) and our FedPAGE (see Section 6.2). The experiments show that FedPAGE always converges much faster than FedAvg, and at least as fast as SCAFFOLD (usually much better than SCAFFOLD), confirming the practical superiority of FedPAGE.

## 1.2 RELATED WORKS

Optimization algorithms for federated learning have a close relationship with the algorithms designed for standard finite-sum problem $\min_x \frac{1}{n} \sum_{i=1}^{n} f_i(x)$. In the federated learning setting, we can think of the loss function of the data on a single client as function $f_i$, and the optimization problem becomes a finite-sum problem. The SGD is perhaps the most famous algorithm for solving the finite-sum problem, and in one variant or another, it is widely applied in training deep neural networks. However, the convergence rates of plain SGD in the convex and nonconvex settings are not optimal. This motivated a feverish research activity in the optimization and machine learning communities over the last decade, and these efforts led to theoretically and practically improved variants of SGD, such as SVRG, SAGA, SARAH, SPIDER, and PAGE (Johnson and Zhang, 2013; Defazio et al., 2014; Nguyen et al., 2017; Fang et al., 2018; Li et al., 2021) and many of their variants possibly with acceleration/momentum (Allen-Zhu, 2017; Lan and Zhou, 2018; Lei et al., 2017; Li and Li, 2018; Zhou et al., 2018; Wang et al., 2018; Kovalev et al., 2020; Ge et al., 2019; Li, 2019; Lan et al., 2019; Li and Li, 2020; Li, 2021a).

Table 1: Number of communication rounds for finding an $\epsilon$-solution of federated convex and non-convex problems (1), where $\mathbb{E}f(x) - f^* \leq \epsilon$ for convex setting and $\mathbb{E}\|\nabla f(x)\|_2 \leq \epsilon$ for nonconvex setting. In the last column, $(G, B)$-*BGD* means that $\sum_{i=1}^N \|\nabla f_i(x)\|_2^2 \leq G^2 + B^2\|\nabla f(x)\|_2^2$. $(G, 0)$-BGD means $B = 0$, and $(0, B)$-BGD means $G = 0$. *BV* denotes the "Bounded Variance" assumption, i.e., (2), and *Smooth* stands for standard smoothness assumption, e.g., Assumption 1. Other notations (i.e., $N, S, M, K$) are summarized in Table 2.

| Algorithm | Convex setting | Nonconvex setting | Assumption |
|---|---|---|---|
| FedAvg (Yu et al., 2019) | — | $\frac{G^2 NK}{\epsilon^2} + \frac{\sigma^2}{NK\epsilon^4}$ | Smooth, BV, $(G, 0)$-BGD |
| FedAvg (Karimireddy et al., 2020) | $\frac{G^2}{S\epsilon^2} + \frac{G}{\epsilon^{3/2}} + \frac{B^2}{\epsilon} + \frac{\sigma^2}{SK\epsilon^2}$ | $\frac{G^2}{S\epsilon^4} + \frac{G}{\epsilon^3} + \frac{B^2}{\epsilon^2} + \frac{\sigma^2}{SK\epsilon^4}$ | Smooth, BV, $(G, B)$-BGD |
| FedProx (Sahu et al., 2018) | $\frac{B^2}{\epsilon}$ | — | Smooth, $S = N$, $(0, B)$-BGD |
| VRL-SGD (Liang et al., 2019) | — | $\frac{N}{\epsilon^2} + \frac{N\sigma^2}{K\epsilon^4}$ | Smooth, BV, $S = N$ |
| S-Local-SVRG (Gorbunov et al., 2020) | $\frac{M^{1/3}/K^{1/3}+\sqrt{M/NK^2}}{\epsilon}$ [1] | — | Smooth, (BV), $S = N$, $K \leq M$ |
| SCAFFOLD (Karimireddy et al., 2020) | $\frac{N}{S\epsilon} + \frac{\sigma^2}{SK\epsilon^2}$ | $\frac{N^{2/3}}{S^{2/3}\epsilon^2} + \frac{\sigma^2}{SK\epsilon^4}$ | Smooth, BV |
| FedPAGE (this paper) | $\frac{N^{3/4}}{S\epsilon}$ [2] | $\frac{N^{1/2}+S}{S\epsilon^2}$ | Smooth, (BV) [3] |

However, the above well-studied finite-sum problem is not equivalent to the federated learning problem (1) as one needs to account for the communication, which forms the main bottleneck. As we discussed before, there are at last two sets of ideas for solving this problem: communication compression, and local computation. There are lots of works belonging to these two categories. In particular, for the first category, the current state-of-the-art results in strongly convex, convex, and nonconvex settings are given by Li et al. (2020); Li and Richtárik (2021a); Gorbunov et al. (2021), respectively. For the second category, local methods such as FedAvg (McMahan et al., 2017) and SCAFFOLD (Karimireddy et al., 2020) perform multiple local update steps in each communication round in the hope that these are useful to decrease the number of communication rounds needed to train the model. In this paper, we provide new state-of-the-art results of local methods for both federated convex and nonconvex settings, which significantly improves the previous best-known results of SCAFFOLD (Karimireddy et al., 2020) (See Table 1).

## 2 SETUP AND NOTATION

We formalize the problem as minimizing a finite-sum functions:

$$\min_{x \in \mathbb{R}^d}\left\{ f(x) := \frac{1}{N}\sum_{i=1}^N f_i(x) \right\}, \text{where } f_i(x) := \frac{1}{M}\sum_{i=1}^M f_{i,j}(x). \tag{1}$$

In this formulation, each function $f_i(\cdot)$ stands for the loss function with respect to the data stored on client/device/machine $i$, and each function $f_{i,j}(\cdot)$ stands for the loss function with respect to the $j$-th data on client $i$. Besides, we assume that the minimum of $f$ exists, and we use $f^*$ and $x^*$ to denote the minimum of function $f$ and the optimal point respectively.

---

[1] We point out that S-Local-SVRG (Gorbunov et al., 2020) only considered the case where $S = N$ and $K \leq M$, i.e., the sampled clients $S$ always is the whole set of clients $N$ for all communication rounds. As a result, the total communication complexity (i.e., number of rounds $\times$ communicated clients $S$ in each round) of S-Local-SVRG is $O((NM^{1/3}/K^{1/3} + \sqrt{NM/K^2})/\epsilon)$ (note that here $K \leq M$), which is worse than $O(N/\epsilon)$ of SCAFFOLD (Karimireddy et al., 2020) and $O(N^{3/4}/\epsilon)$ of our FedPAGE.

[2] In the convex setting, we state the result of FedPAGE in the case $S \leq \sqrt{N}$ (typical in practice) for simple presentation, where $N$ is the total number of clients and $S$ is the number of sampled subset clients in each communication round (see Table 2). Please see Theorem 2 for other results of FedPAGE in the cases $S > \sqrt{N}$.

[3] FedPAGE also works under the BV assumption by using moderate minibatches for local clients, and more importantly the number of communication rounds of FedPAGE still remains the same as in the last row of Table 1 for both convex (see Theorem 2) and nonconvex (see Theorem 1) settings.

Table 2: Summary of notation used in this paper

| | |
|---|---|
| $N, S, i$ | total number, sampled number, and index of clients |
| $M$ | total number of data in each client |
| $R, r$ | total number and index of communication rounds |
| $K, k$ | total number and index of local update steps |
| $x^r$ | model parameters before round $r$ |
| $g^r$ | server update within round $r$ |
| $y_{i,k}^r$ | $i$-th client's model in round $r$ before local step $k$ |
| $g_{i,k}^r$ | $i$-th client's update in round $r$ within local step $k$ |
| $\nabla_{\mathcal{I}} f_i(x)$ | estimator of $\nabla f_i(x)$ using a sampled minibatch $\mathcal{I}$ |
| | $\nabla_{\mathcal{I}} f_i(x) = 1/|\mathcal{I}| \sum_{j \in \mathcal{I}} \nabla f_{i,j}(x)$ |

We will use $[n]$ to denote the set $\{1, 2, \ldots, n\}$, $\|\cdot\|$ to denote the Euclidean norm for a vector, and $\langle u, v \rangle$ to denote the inner product of $u$ and $v$. We use $O(\cdot)$ and $\Omega(\cdot)$ to hide the absolute constants.

In this paper, we consider two cases: nonconvex case and convex case. In the nonconvex case, each individual function $f_i$ and the average function $f$ can be nonconvex, and we assume that the functions $\{f_{i,j}\}_{i \in [N], j \in [M]}$ are $L$-smooth.

**Assumption 1** ($L$-smoothness). *All functions $f_{i,j} : \mathbb{R}^d \to \mathbb{R}$ for all $i \in [N], j \in [M]$ are $L$-smooth. That is, there exists $L \geq 0$ such that for all $x_1, x_2 \in \mathbb{R}^d$ and all $i \in [N], j \in [M]$,*

$$\|\nabla f_{i,j}(x_1) - \nabla f_{i,j}(x_2)\| \leq L\|x_1 - x_2\|.$$

If the functions $\{f_{i,j}\}_{i \in [N], j \in [M]}$ are $L$-smooth, we can conclude that functions $\{f_i\}$ are also $L$-smooth and function $f$ is $L$-smooth. In this nonconvex setting, the optimization algorithm aims to find a point such that the expectation of the gradient norm is small enough: $\mathbb{E}\|\nabla f(x)\|_2 \leq \epsilon$.

Then, in the convex case, each *individual function $f_i$* can be *nonconvex*, but we require that the *average function $f$* to be *convex*. We also assume that the functions $\{f_{i,j}\}$ are $L$-smooth (Assumption 1). Under this convex setting, the algorithm will find a point such that the expectation of the function value is close to the minimum: $\mathbb{E}f(x) - f^* \leq \epsilon$.

If the number of data on a single client is very large and one cannot compute the local full gradients of clients, one needs the following assumption in which the gradient variance on each client is bounded.

**Assumption 2** (Bounded Variance). *There exists $\sigma \geq 0$ such that for any client $i \in [N]$ and $x \in \mathbb{R}^d$,*

$$\frac{1}{M} \sum_{j=1}^{M} \|\nabla f_{i,j}(x) - \nabla f_i(x)\|_2^2 \leq \sigma^2. \tag{2}$$

## 3 THE FedPAGE ALGORITHM

In this section, we introduce our FedPAGE algorithm. To some extent, our FedPAGE algorithm is the local version of PAGE (Li et al., 2021): when the clients communicate with the server, FedPAGE behaves similar to PAGE, and when the clients update the model locally, each client updates several steps. If we set the number of local updates to one, FedPAGE reduces to the original PAGE algorithm.

Our FedPAGE algorithm is given in Algorithm 1. There are two cases in each round $r$: 1) with probability $p_r$ (typically very small), the server communicates with all clients in order to get a more accurate gradient of function $f$ (Line 3–9); 2) with probability $1 - p_r$, the server communicates with a subset of clients with size $S$ and the local clients perform $K$ local steps (Line 10–24).

For Case 1), the server broadcasts the current model parameters $x^r$ to all of the clients. Then, each client computes the estimator $\nabla_{\mathcal{I}_1} f_i(x^r)$ of the gradient $\nabla f_i(x^r)$ and sends to the server. The estimator takes $b_1$ minibatch samples ($|\mathcal{I}_1| = b_1$) to estimate the gradient of $f_i$ and different clients sample different sets $\mathcal{I}_1$. Here, we want $\nabla_{\mathcal{I}_1} f_i(x^r)$ to be as closed to $\nabla f_i(x^r)$ as possible, choosing a moderate size $b_1$ usually is enough. The server average all of the gradient and get the averaged gradient $g^r = \frac{1}{N} \sum_{i \in [N]} \nabla_{\mathcal{I}_1} f_i(x^r)$ and takes a step with global step size $\eta_g$ (see Line 9, 26).

---

**Algorithm 1** FedPAGE

---

**server input:** initial point $x^0$, global step size $\eta_g$, probabilities $\{p_r\}$, sampled clients size $S$
**client $i$'s input:** local step size $\eta_l$, minibatch sizes $b_1, b_2, b_3$

  1: **for** $r = 0, 1, 2, \ldots, R$ **do**
  2:       sample $q \sim \text{Bernoulli}(p_r)$
  3:       **if** $q = 1$ **then**
  4:             clients $S^r = [N]$, communicate $x^r$ to all $i \in S^r$
  5:             **on client** $i \in S^r$ **in parallel do**
  6:                 uniformly sample minibatch $\mathcal{I}_1 \subset [M]$ with size $b_1$
  7:                 compute the gradient estimator $g_i^r \leftarrow \nabla_{\mathcal{I}_1} f_i(x^r)$
  8:             **end on client**
  9:             $g^r \leftarrow \frac{1}{N} \sum_{i \in [N]} g_i^r$
 10:       **else**
 11:             sample clients $S^r \subseteq [N]$ with size $S$, communicate $(x^r, x^{r-1}, g^{r-1})$ to all $i \in S^r$
 12:             **on client** $i \in S^r$ **in parallel do**
 13:                 $y_{i,0}^r \leftarrow x^r$
 14:                 uniformly sample minibatch $\mathcal{I}_2 \subset [M]$ with size $b_2$
 15:                 $g_{i,0}^r \leftarrow \nabla_{\mathcal{I}_2} f_i(x^r) - \nabla_{\mathcal{I}_2} f_i(x^{r-1}) + g^{r-1}$
 16:                 $y_{i,1}^r \leftarrow y_{i,0}^r - \eta_l g_{i,0}^r$
 17:                 **for** $k = 1, 2, \ldots, K - 1$ **do**
 18:                     uniformly sample minibatch $\mathcal{I}_3 \subset [M]$ with size $b_3$
 19:                     $g_{i,k}^r \leftarrow \nabla_{\mathcal{I}_3} f_i(y_{i,k}^r) - \nabla_{\mathcal{I}_3} f_i(y_{i,k-1}^r) + g_{i,k-1}^r$
 20:                     $y_{i,k+1}^r \leftarrow y_{i,k}^r - \eta_l g_{i,k}^r$
 21:                 **end for**
 22:                 $\Delta y_i^r \leftarrow x^r - y_{i,K}^r$
 23:             **end on client**
 24:             $g^r \leftarrow \frac{1}{K\eta_l S} \sum_{i \in S^r} \Delta y_i^r$
 25:       **end if**
 26:       $x^{r+1} \leftarrow x^r - \eta_g g^r$
 27: **end for**

---

For Case 2), the server first broadcasts $(x^r, x^{r-1}, g^{r-1})$ to the sampled subset clients $S^r$, and the clients initialize $y_{i,0}^r \leftarrow x^r$. Here, $y_{i,0}^r$ is $i$-th client's model in round $r$ before local step $k$, and $g_{i,k}^r$ denotes $i$-th client's gradient estimator for step $k$ in round $r$. Then for the first local step of client $i$, the local gradient estimator is computed in Line 15 as

$$g_{i,0}^r \leftarrow \nabla_{\mathcal{I}_2} f_i(x^r) - \nabla_{\mathcal{I}_2} f_i(x^{r-1}) + g^{r-1},$$

where $\nabla_{\mathcal{I}_2} f_i(\cdot)$ is the gradient estimator of $\nabla f_i(\cdot)$ with minibatch size $b_2$. Here, we also want $\nabla_{\mathcal{I}_2} f_i(\cdot)$ to be as closed to $\nabla f_i(\cdot)$ as possible, similarly choosing a moderate size $b_2$ usually is enough. This update rule is similar to PAGE (Li et al., 2021) and in particular if the local steps $K = 1$, our FedPAGE algorithm reduces to PAGE.

For client $i$'s local step $k$ such that $1 \le k \le K - 1$, the gradient estimator is computed in Line 19 as

$$g_{i,k}^r \leftarrow \nabla_{\mathcal{I}_3} f_i(y_{i,k}^r) - \nabla_{\mathcal{I}_3} f_i(y_{i,k-1}^r) + g_{i,k-1}^r.$$

Here $\mathcal{I}_3$ is a minibatch of functions with size $b_3$ that we used to compute the gradient estimator $g_{i,k}^r$. Different from the previous gradient estimators using minibatches with size $b_1$ and $b_2$, here we want $b_3$ to be small enough to reduce the computation cost as there are $K$ local steps (Line 17–21). In particular, we can choose $b_3 = 1$, i.e., just sample an index $j$ from $[M]$ and the estimator becomes

$$g_{i,k}^r \leftarrow (\nabla f_{i,j}(y_{i,k}^r) - \nabla f_{i,j}(y_{i,k-1}^r)) + g_{i,k-1}^r.$$

The local model update is given by $y_{i,k+1}^r = y_{i,k}^r - \eta_l g_{i,k}^r$ where $\eta_l$ is the local step size. After $K$ local steps, client $i$ computes the local changes $\Delta y_i^r = y_{i,K}^r - x^r$ within round $r$ and sends back to the server. After receiving the local changes $\Delta y_i^r$ for the selected clients $i \in S^r$, the server computes the average gradient estimator on these selected clients in Line 24 as $g^r = \frac{1}{K\eta_l S} \sum_{i \in S^r} \Delta y_i^r$.

After obtaining the gradient estimator $g^r$ (in Line 9 or 24), the server updates the model using a global step size $\eta_g$ in Line 26 as $x^{r+1} = x^r - \eta_g g^r$.

The intuition of FedPAGE works as follow: when the local step size $\eta_l$ is not too large, we can expect that the local model updates are close to the original model, that is $y^r_{i,k} \approx x^r$, and the gradient is also close to each other, $\nabla f_{i,j}(y^r_{i,k}) \approx \nabla f_{i,j}(x^r)$. Then each local gradient estimator $g^r_{i,k}$ is close to

$$g^r_{i,k} = (\nabla f_{i,j}(y^r_{i,k}) - \nabla f_{i,j}(y^r_{i,k-1})) + g^r_{i,k-1} \approx \nabla f_{i,j}(x^r) - \nabla f_{i,j}(x^r) + g^r_{i,k-1} = g^r_{i,k-1} = g^r_{i,0},$$

and the aggregated global gradient estimator $g^r$ is close to

$$g^r \approx \frac{1}{S} \sum_{i \in S^r} \left( \nabla f_i(x^r) - \nabla f_i(x^{r-1}) + g^{r-1} \right).$$

This biased recursive gradient estimator $g^r$ is similar to the gradient estimator in SARAH (Nguyen et al., 2017) or PAGE (Li et al., 2021), and thus the performance of FedPAGE in terms of communication rounds should be similar to the optimal convergence results of PAGE (Li et al., 2021).

## 4    FedPAGE IN NONCONVEX SETTING

In this section, we provide the general result of our FedPAGE with any local steps $K \geq 1$ in the nonconvex setting. Here we assume that the functions $\{f_{i,j}\}$ are $L$-smooth (Assumption 1), and we obtain the following theorem.

**Theorem 1** (Convergence of FedPAGE in nonconvex setting). *Under Assumption 1 (and Assumption 2), if we choose the sampling probability $p_r \equiv p = \frac{S}{N}$ for every $r \geq 1$ and $p_0 = 1$, the minibatch sizes $b_1 = \min\{M, \frac{24\sigma^2}{S\epsilon^2}\}, b_2 = \min\{M, \frac{48\sigma^2}{pS\epsilon^2}\}$, and the global and local step sizes*

$$\eta_g \leq \frac{1}{L\left(1 + \sqrt{\frac{3(1-p/3)}{2pS}}\right)}, \qquad \eta_l \leq \frac{\sqrt{2}p}{24\sqrt{S}KL},$$

*then FedPAGE will find a point $x$ such that $\mathbb{E}\|\nabla f(x)\|_2 \leq \epsilon$ within the following number of communication rounds:*

$$R = O\left(\frac{L(\sqrt{N} + S)}{S\epsilon^2}\right).$$

Now we compare the communication cost of FedPAGE (Theorem 1) with previous state-of-the-art SCAFFOLD (Karimireddy et al., 2020). The number of communication round for SCAFFOLD to find a point $x$ such that $\mathbb{E}\|\nabla f(x)\|_2 \leq \epsilon$ (the original SCAFFOLD (Karimireddy et al., 2020) uses $\mathbb{E}\|\nabla f(x)\|_2^2 \leq \epsilon$) is bounded by

$$R_{\text{SCAFFOLD}} = O\left((N/S)^{2/3} L/\epsilon^2\right). \tag{3}$$

Beyond the number of communication rounds in FedPAGE and SCAFFOLD, we also need to compare the communication cost during each round (i.e., number of clients communicated with the server in the round). For our FedPAGE, in each round, with probability $p = \frac{S}{N}$, the server communicates with all clients $N$, and with probability $1 - p$, the server communicates with a sampled subset clients with size $S$, and the communicated clients within each round is $\frac{S}{N} \times N + (1 - \frac{S}{N}) \times S < 2S$ in expectation. For SCAFFOLD, in each round, the server communicates with $S$ sampled clients. Thus the communication cost for each round is the same $O(S)$ for both FedPAGE and SCAFFOLD. As a result, to compare the communication complexity of FedPAGE and SCAFFOLD, it is equivalent to compare the number of communication rounds. According to (3) and Theorem 1 (e.g., with sampled clients $S \leq \sqrt{N}$), then the communication rounds of FedPAGE is smaller than previous state-of-the-art SCAFFOLD by a factor of $N^{1/6}S^{1/3}$. Also note that the number of clients $N$ is usually very large in the federated learning problems.

## 5 FedPAGE IN CONVEX SETTING

In this section, we show the convergence results of FedPAGE in the convex setting. Here the algorithms aim to find a point $x$ such that $\mathbb{E}f(x) - f^* \leq \epsilon$ for convex case instead of $\mathbb{E}\|\nabla f(x)\|_2 \leq \epsilon$ for nonconvex case. In this part, we assume that $f$ is convex and the functions $\{f_{i,j}\}$ satisfy $L$-smoothness assumption (Assumption 1).

**Theorem 2** (Convergence of FedPAGE in convex setting). *Under Assumption 1 (and Assumption 2), if we choose the sampling probability $p_r = p = \frac{S}{N}$ for every $r \geq 1$ and $p_0 = 1$, the minibatch sizes $b_1 = \min\{M, \frac{24\sigma^2}{p^{1/2}\sqrt{S}\epsilon}\}, b_2 = \min\{M, \frac{48\sigma^2}{p^{3/2}\sqrt{S}\epsilon}\}$, and the global and local step size*

$$\eta_g = \Theta\left(\frac{(S + N^{3/4})\frac{S}{N}}{L(S + \sqrt{N})}\right), \qquad \eta_l = O\left(\frac{S}{N^{5/4}KL_c\sqrt{T}}\right),$$

*then* FedPAGE *satisfies*

$$\frac{1}{R}\sum_{r=0}^{R-1} \mathbb{E}[f(x^{r+1}) - f(x^*)] \leq \begin{cases} O\left(\frac{N^{3/4}L}{SR} + \epsilon\right), \text{if } S \leq \sqrt{N} \\ O\left(\frac{N^{1/4}L}{R} + \epsilon\right), \text{if } \sqrt{N} < S \leq N^{3/4} \\ O\left(\frac{NL}{SR} + \epsilon\right), \text{if } N^{3/4} < S \end{cases}.$$

As we discussed before, the expected communication cost of FedPAGE is the same as SCAFFOLD in each communication round. Then if the sampled clients $S \leq \sqrt{N}$, FedPAGE can find a solution $x$ such that $\mathbb{E}f(x) - f(x^*) \leq \epsilon$ within $O(\frac{N^{3/4}L}{S\epsilon})$ number of communication rounds, improving the previous state-of-the-art $O(\frac{N}{S\epsilon})$ of SCAFFOLD (Karimireddy et al., 2020) by a large factor of $N^{1/4}$. Recall that $N$ denotes the total number of clients.

## 6 NUMERICAL EXPERIMENTS

In this section, we present our numerical experiments. We conducted two experiments: the first shows the effectiveness of the local steps (Section 6.1), and the second compares FedPAGE with SCAFFOLD and FedAvg (Section 6.2). Before we present the results of these two experiments, we first state the general experiment setups. The detailed experiment setups are deferred to the appendix.

**Experiment setup** We run experiments on two nonconvex problems used in e.g. (Wang et al., 2018; Li and Richtárik, 2021b): robust linear regression and logistic regression with nonconvex regularizer. The standard datasets a9a (32,561 samples) and w8a (49,749 samples) are downloaded from LIBSVM (Chang and Lin, 2011). The objective function for robust linear regression is

$$f(x) = \frac{1}{n}\sum_{i=1}^{n} \ell(x^T a_i - b_i),$$

where $\ell(t) = \log(1 + \frac{t^2}{2})$. Here $b_i \in \{\pm 1\}$ is a binary label.

The objective function for logistic regression with nonconvex regularizer is

$$f(x) = \frac{1}{n}\sum_{i=1}^{n} \log\left(1 + \exp(-b_i x^T a_i)\right) + \alpha \sum_{j=1}^{d} \frac{x_j^2}{1+x_j^2}.$$

Here, the last term is the regularizer term and we set $\alpha = 0.1$.

Besides, different algorithms have different definitions of the local step size and global step size, thus we compare these algorithms with the 'effective step size' $\tilde{\eta}$. Here for FedPAGE, the effective step size is just the global step size $\tilde{\eta} = \eta_g$, and for SCAFFOLD and FedAvg, the effective step size is defined as $\tilde{\eta} = K\eta_g\eta_l$. We run experiments with $\tilde{\eta} = 0.1, 0.03, 0.01$. If the effective step size is larger, the algorithms may diverge. Also note that although we compare these algorithms with the same effective step size, FedPAGE can use a larger step size from our theoretical results. Finally we select the total number of communication rounds such that the algorithms converge or we can distinguish their performance difference.

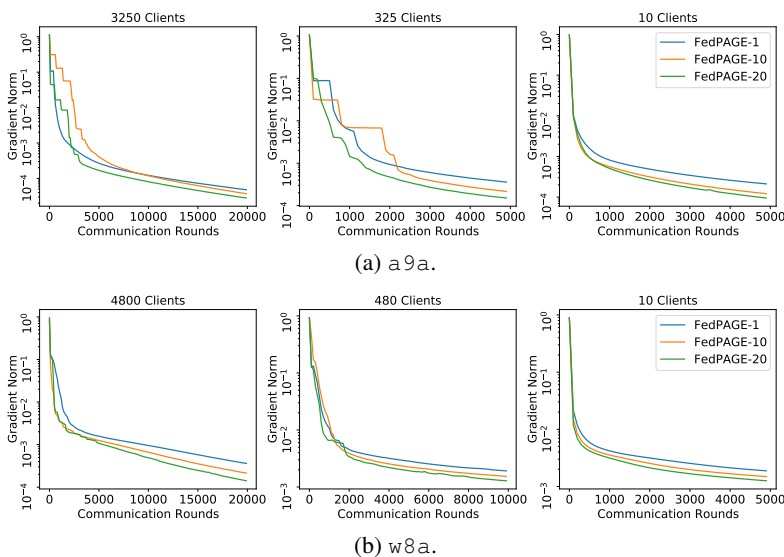

Figure 1: FedPAGE with different number of local steps on different datasets.

## 6.1 EFFECTIVENESS OF LOCAL STEPS

In this experiment we compare the convergence performance of FedPAGE using different number of local update steps: $K = 1, 10, 20$ (see Line 17 of Algorithm 1). FedPAGE-1 means that the number of local step $K = 1$, which reduces to the original PAGE (Li et al., 2021), and FedPAGE-10 and FedPAGE-20 represent FedPAGE with 10 and 20 local steps respectively.

The experimental results are presented in Figure 1. Figure 1a shows the robust linear regression results of FedPAGE using different number of local steps $K = 1, 10, 20$ on a9a dataset, and Figure 1b shows the result on w8a dataset.

**Local steps speed up the convergence rate**  The experimental results in Figure 1 show that the multiple local steps of FedPAGE can speed up the convergence in terms of the communication rounds. Although there are some fluctuations when the number of communication round is not large (early-stage), FedPAGE-10 and FedPAGE-20 outperform FedPAGE-1 in the end.

**Algorithm with multiple local steps can choose a larger effective step size**  From our hyperparameter optimization results, we also find that FedPAGE with multiple local steps can choose a larger effective step size ($\eta_g$ in FedPAGE). On a9a dataset, when there are 3250 clients, the effective step size for FedPAGE-1, FedPAGE-10, and FedPAGE-20 are optimized to be $0.3, 0.4, 0.4$ respectively; when there are 325 clients, the effective step size for FedPAGE-1, FedPAGE-10, and FedPAGE-20 are optimized to be $0.2, 0.4, 0.5$; when there are 10 clients, the effective step size for FedPAGE-1, FedPAGE-10, and FedPAGE-20 are optimized to be $0.3, 0.5, 0.6$. The experiments on w8a dataset also support this finding.

## 6.2 COMPARISON WITH PREVIOUS METHODS

Now, we compare our FedPAGE with two other methods: SCAFFOLD (Karimireddy et al., 2020) and FedAvg (McMahan et al., 2017). The experimental results are presented in Figure 2 and 3. We plot the gradient norm versus the number of communication rounds. Figures 2a, 2b, 3a, and 3b show the performance of each algorithm using different objective functions and different datasets.

**Performance of different methods**  The experiments show that FedPAGE $\geq$ SCAFFOLD $>$ FedAvg. Among all the cases, under the same effective step size, we find that both FedPAGE and SCAFFOLD converge faster than FedAvg. FedPAGE converges at least as fast as SCAFFOLD, and in most of the cases FedPAGE converges faster than SCAFFOLD.

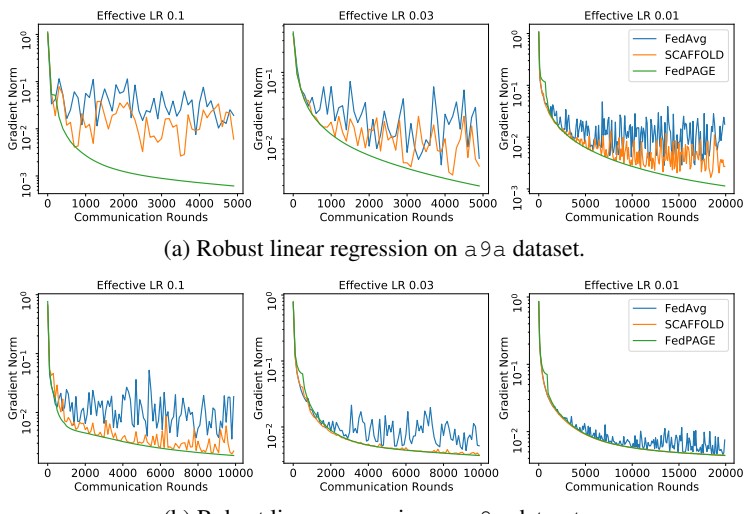

(a) Robust linear regression on `a9a` dataset.

(b) Robust linear regression on `w8a` dataset.

Figure 2: Comparison of different methods with robust linear regression.

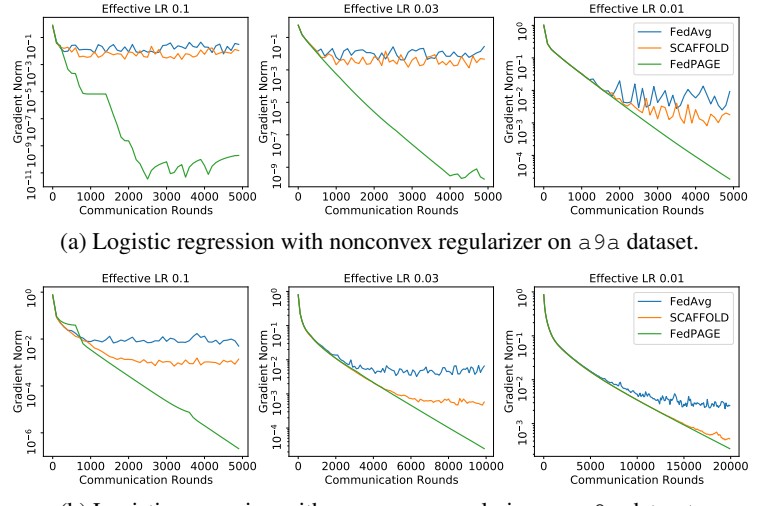

(a) Logistic regression with nonconvex regularizer on `a9a` dataset.

(b) Logistic regression with nonconvex regularizer on `w8a` dataset.

Figure 3: Comparison of different methods with logistic regression with nonconvex regularizer.

**Larger effective step size converges faster** The experiments also show that a larger effective step size leads to a faster convergence as long as the algorithm converges. Note that FedPAGE can use a larger step size with theoretical guarantee compared with SCAFFOLD, if we choose the same parameters of the objective function (e.g. the same smoothness constant) and use the step size with theoretical guarantees, FedPAGE converges faster than SCAFFOLD than FedAvg.

## 7 CONCLUSION

In this paper, we propose a new federated learning algorithm, FedPAGE, providing much better state-of-the-art communication complexity for both federated convex and nonconvex optimization. We also conduct several numerical experiments showing the effectiveness of multiple local update steps in FedPAGE and verifying the practical superiority of FedPAGE over other classical methods.

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

# A  MORE EXPERIMENTS

In this section we describe the detailed experiment setting for Section 6 and present more numerical experiments. We perform two different experiments: the first is to compare the performance of different algorithm with different number of clients and data on a single client (Section A.2), and the second is to compare different algorithm with local full gradient computations, which shows the limitation of different algorithms (Section A.3).

## A.1  DETAILED EXPERIMENT SETTING IN SECTION 6

**Detailed experiment setting in Section 6.1**  We use the robust linear regression as the objective function. We run experiment on the a9a dataset in which the total number of data samples is 32500 (here we drop the last 61 samples for easy implementation of different number of clients). We choose the number of clients to be $3250, 325, 10$, and the numbers of data on a single client are $10, 100$, and $3250$, respectively. When the number of clients is 3250, we choose $S = 10$, i.e. the server communicates with 10 clients in each communication round, and when the number of clients is 325 or 10, we set $S = 1$. For all settings, we optimize the global step size $\eta_g$ and choose the local step size $\eta_l$ heuristically such that the algorithms converge as fast as possible. For FedPAGE-1 (or PAGE), the local step size does not matter and choosing the optimal global step size achieves its best convergence rate, however for FedPAGE-10 and FedPAGE-20, choosing $\eta_g, \eta_l$ with some heuristics does not guarantee the best performance. We also perform the similar experiments on another dataset w8a.

**Detailed experiment setting in Section 6.2**  For the experiments with a9a dataset, we omit the last 61 samples and set the number of clients to be 3250, and for experiments with w8a, we omit the last 1749 samples and there are 4800 clients in total. Here we omit the samples because it is more convenient to change the number of clients. We let each 'client' contains 10 samples from the datasets. For SCAFFOLD and FedAvg, in each communication round, the server will communicate with 20 clients ($S = 20$ in their algorithms). For FedPAGE, we set $S = 10$ because FedPAGE will communicate with all clients with probability $\frac{S}{N}$ and the expected communication for all three algorithms in each round are almost the same. We choose the local steps of all these three methods to be 10. For FedPAGE, we choose the minibatch size $b_3 = 1$ and for SCAFFOLD and FedAvg, we choose the minibatch size that estimate the local full gradient to be 4. In this way, the local computations are nearly the same for all methods.

## A.2  COMPARISON OF DIFFERENT METHODS WITH DIFFERENT NUMBER OF CLIENTS

### A.2.1  EXPERIMENT SETUP

In previous Section 6, we compare different methods with a large number of clients (on a9a dataset, there are 3250 clients, and on w8a dataset, there are 4800 clients). In this experiment, we vary the number of clients and compare the performance of FedPAGE, SCAFFOLD, and FedAvg.

For the number of clients, we choose the number of clients to be 325 and 10, and the number of data on a single client are 100 and 3250. We choose the number of local steps to be 10 for all three methods. When the number of clients are 325 and 10, we set $S = 1$ for FedPAGE and $S = 2$ for SCAFFOLD and FedAvg, making the communication cost in each round to be nearly the same. We set FedPAGE to compute the full local gradient for the first local step, and choose only one sample to estimate the gradient for the following local steps. For SCAFFOLD and FedAvg, we set the minibatch size estimating the local full gradient to be 22 when the number of client is 325 and 652 when the number of client is 10. When the number of client is 325, there are 100 data on a single client. FedPAGE need to compute two full gradient at the beginning of each local computations, costing 200 number of gradient computations. Then it needs to compute two gradient (the gradient of a same sample at different points), and it cost about 220 gradient computations in total. Choosing the minibatch size to be 22 in SCAFFOLD and FedAvg makes the local computations nearly the same, because SCAFFOLD and FedAvg use the same minibatch size in every local step. When the number of client is 10, the minibatch size for SCAFFOLD and FedAvg can be computed as $3250 \times 2/10 + 2 = 652$. This makes the local computations of these three algorithms to be nearly the same.

For the step sizes, we choose the effective step sizes to be $0.1$, $0.03$, and $0.01$.

### A.2.2 EXPERIMENT RESULTS

The experimental results are presented in Figure 4. Figure 4b and 4c shows the experiment results with 325 clients and 10 clients on `a9a` dataset. We also include Figure 4a (i.e., Figure 2a in Section 6.2) with 3250 clients for better comparison. Similar to the experimental results in Section 6, Figure 4 also demonstrates that `FedPAGE` typically converges faster than `SCAFFOLD` faster than `FedAvg`.

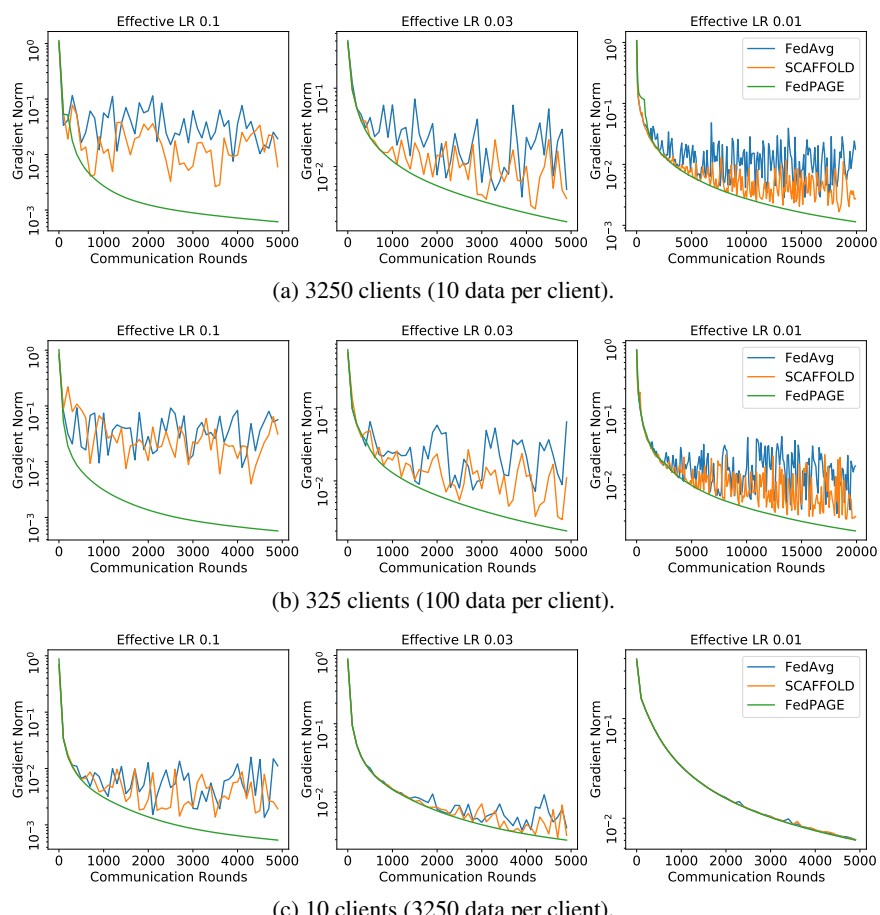

(a) 3250 clients (10 data per client).

(b) 325 clients (100 data per client).

(c) 10 clients (3250 data per client).

Figure 4: Experiment results of different methods with different number of clients.

### A.3 COMPARISON OF DIFFERENT METHODS WITH LOCAL FULL GRADIENT COMPUTATION

### A.3.1 EXPERIMENT SETUP

In this section, we design another experiment to observe the performance limitation of `FedPAGE`, `SCAFFOLD`, and `FedAvg`. We substitute all the steps that use a minibatch to estimate the local full gradient to the actual full gradient computation. In `FedPAGE`, we choose $b_3 = 1$ in the previous experiments and now we set $b_3 = M$, the number of data on a single client. We also choose $b_1 = b_2 = M$. We denote the resulting algorithm `FedPAGE-Full`. Similarly, for `SCAFFOLD` and `FedAvg`, they choose a minibatch to estimate the local full gradient, and now we change them to computing the local full gradient, i.e., $b = M$. We denote the resulting algorithms as `SCAFFOLD-Full` and `FedAvg-Full`.

We then compare four different methods: `FedPAGE`, `FedPAGE-Full`, `SCAFFOLD-Full`, and `FedAvg-Full`. We perform the experiments on `a9a` and `w8a` datasets with robust linear regression objective and

logistic regression with nonconvex regularizer objective. We let each 'client' contains 10 samples from the dataset. We set all the algorithm to run with 10 local steps ($K = 10$). We run the experiments with effective step size 0.1, 0.03, and 0.01. For experiment on w8a dataset with logistic regression with nonconvex regularizer, we also test the algorithms with effective step size $0.3$. For FedPAGE and FedPAGE-Full, we set $S = 10$ and for SCAFFOLD-Full and FedAvg-Full, we set $S = 20$ to make the communication cost in each round to be nearly the same.

### A.3.2 EXPERIMENT RESULTS

The experimental results are presented in Figure 5 and 6. Figure 5a, 5b, 6a, and 6b show the results of different methods on different problems and different datasets as stated in their captions.

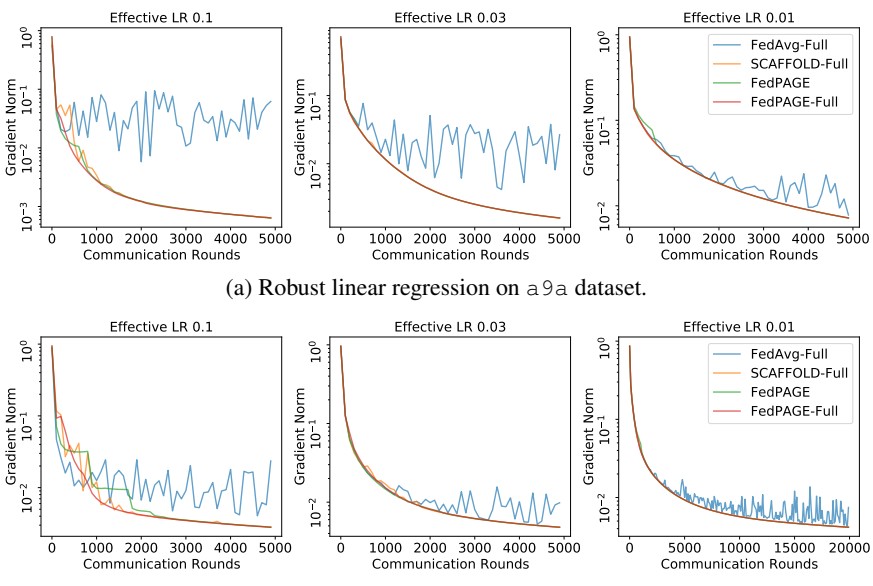

(a) Robust linear regression on a9a dataset.

(b) Robust linear regression on w8a dataset.

Figure 5: Comparison of different methods with robust linear regression.

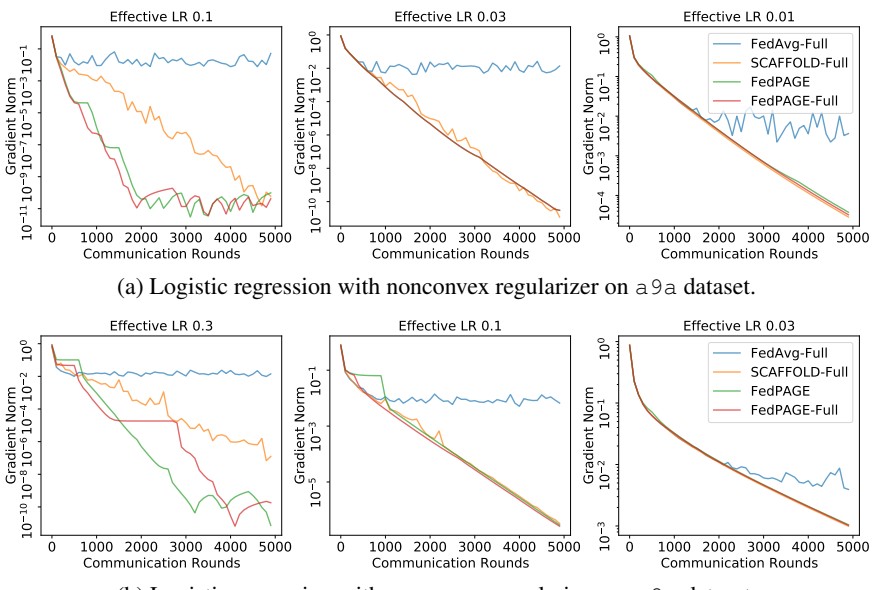

(a) Logistic regression with nonconvex regularizer on a9a dataset.

(b) Logistic regression with nonconvex regularizer on w8a dataset.

Figure 6: Comparison of different methods with logistic regression with nonconvex regularizer.

FedPAGE ≈ FedPAGE-Full    First, the experiments show that the convergence performance of FedPAGE and FedPAGE-Full are nearly the same under the same effective step size. Although there are some fluctuations in the convergence process, the fluctuations are not large enough to conclude any difference between the convergence speed of FedPAGE and FedPAGE-Full.

FedPAGE-Full ≥ SCAFFOLD-Full > FedAvg-Full    Next, the experiments show that FedPAGE-Full converges at least as fast as (usually outperforms) SCAFFOLD-Full and both of them converge faster than FedAvg-Full in all cases. Using the robust linear regression objective in Figure 5, FedPAGE-Full and SCAFFOLD-Full converges nearly at the same speed, but in the experiments with logistic regression with nonconvex regularizer in Figure 6, FedPAGE-Full usually outperforms SCAFFOLD-Full especially when the effective step size is large. From the experiments, FedPAGE either has faster convergence performance under the same local computation cost, or can use less local computational resources and achieve the same or even better performance.

## B    GRADIENT COMPLEXITY OF DIFFERENT METHODS

In previous Table 1, we show the number of communication rounds of different methods. In this section, we compare the gradient complexity among different methods. Table 3 summarizes the gradient complexity per client of different methods under different assumptions.

For SCAFFOLD, in each communication round, $S$ selected clients need to perform $K$ local steps, and the gradient computations of local client is the number of communication round times $SK/N$. For FedPAGE, in each communication round, $S$ selected clients need to first compute two full/moderate minibatch gradients, and then performs $K$ local steps computing only $O(1)$ number of gradient in each step. The gradient complexity per client of FedPAGE is the number of communication round times $S(M + K)/N$. In the BV setting, the full gradient may not be available, then FedPAGE uses a moderate minibatch gradient to estimate the full gradient, and one only needs to change $M$ to the moderate minibatch size in order to obtain the corresponding gradient complexity (See the last two rows in Table 3).

In particular, if the number of data on a single client/device is not very large ($M$ is not very large), one can choose $K$ such that $M + K = O(K)$. Then the number of gradient computed by FedPAGE during a communication round is similar to that computed by SCAFFOLD, and also the number of communication rounds of FedPAGE is much smaller than that of SCAFFOLD regardless of settings (see Table 1). As a result, FedPAGE is strictly much better than SCAFFOLD in terms of both communication complexity and computation complexity, both by a factor of $N^{1/4}$ in the convex setting and $N^{1/6}S^{1/3}$ in the nonconvex setting. Thus, FedPAGE is more suitable for the federated

Table 3: Number of gradient computations per client for finding an $\epsilon$-solution of federated convex and nonconvex problems (1).

| Algorithm | Convex setting | Nonconvex setting | Assumption |
|---|---|---|---|
| FedAvg (Yu et al., 2019) | — | $\frac{G^2NK^2}{\epsilon^2} + \frac{\sigma^2}{N\epsilon^4}$ | Smooth, BV $(G, 0)$-BGD |
| FedAvg (Karimireddy et al., 2020) | $\frac{G^2K}{N\epsilon^2} + \frac{GSK}{N\epsilon^{3/2}} + \frac{B^2SK}{N\epsilon} + \frac{\sigma^2}{N\epsilon^2}$ | $\frac{G^2K}{N\epsilon^4} + \frac{GSK}{N\epsilon^3} + \frac{B^2SK}{N\epsilon^2} + \frac{\sigma^2}{N\epsilon^4}$ | Smooth, BV $(G, B)$-BGD |
| FedProx (Sahu et al., 2018) | $\frac{B^2}{\epsilon}$ | — | Smooth, $S = N$, $(0, B)$-BGD |
| VRL-SGD (Liang et al., 2019) | — | $\frac{NK}{\epsilon^2} + \frac{N\sigma^2}{\epsilon^4}$ | Smooth, BV, $S = N$ |
| S-Local-SVRG (Gorbunov et al., 2020) | $\frac{K+\sqrt{M/N}+M^{1/3}K^{2/3}}{\epsilon}$ | — | Smooth, (BV), $S = N$, $K \leq M$ |
| SCAFFOLD (Karimireddy et al., 2020) | $\frac{K}{\epsilon} + \frac{\sigma^2}{N\epsilon^2}$ | $\frac{S^{1/3}K}{N^{1/3}\epsilon^2} + \frac{\sigma^2}{N\epsilon^4}$ | Smooth, BV |
| FedPAGE (this paper) | $\frac{N^{3/4}}{N\epsilon}(M + K)$ | $\frac{N^{1/2}+S}{N\epsilon^2}(M + K)$ | Smooth |
| FedPAGE (this paper) | $\frac{N^{3/4}}{N\epsilon}\left(\frac{N^{3/2}\sigma^2}{S\epsilon^2} + K\right)$ | $\frac{N^{1/2}+S}{N\epsilon^2}\left(\frac{N\sigma^2}{S\epsilon^2} + K\right)$ | Smooth, BV |

learning tasks that have many devices and each device has limited number of data, e.g. mobile phones.

## C  USEFUL INEQUALITIES

In this part we recall some classical inequalities that helps our derivation.

**Proposition 1.** *Let* $\{\boldsymbol{v}_1, \ldots, \boldsymbol{v}_\tau\}$ *be* $\tau$ *vectors in* $\mathbb{R}^d$. *Then,*

$$\langle \boldsymbol{v}_i, \boldsymbol{v}_j \rangle \leq \frac{c}{2}\|\boldsymbol{u}\|^2 + \frac{1}{2c}\|\boldsymbol{v}\|^2, \forall c > 0. \tag{4}$$

$$\|\boldsymbol{v}_i + \boldsymbol{v}_j\|^2 \leq (1 + \alpha)\|\boldsymbol{v}_i\|^2 + \left(1 + \frac{1}{\alpha}\right)\|\boldsymbol{v}_j\|^2, \forall \alpha > 0. \tag{5}$$

$$\left\|\sum_{i=1}^{\tau} \boldsymbol{v}_i\right\|^2 \leq \tau \sum_{i=1}^{\tau} \|\boldsymbol{v}_i\|^2. \tag{6}$$

$$\tag{7}$$

**Proposition 2.** *If* $X \in \mathbb{R}^d$ *is a random variable, then*

$$\mathbb{E}\|X\|^2 = \mathbb{E}\|X - \mathbb{E}X\|^2 + \|\mathbb{E}X\|^2. \tag{8}$$

*Besides, we have*

$$\mathbb{E}\|X - \mathbb{E}X\|^2 \leq \mathbb{E}\|X\|^2. \tag{9}$$

*If* $X, Y \in \mathbb{R}^d$ *are independent random variables and* $\mathbb{E}Y = \boldsymbol{0}$, *then we have*

$$\mathbb{E}\|X + Y\|^2 = \mathbb{E}\|X\|^2 + \mathbb{E}\|Y\|^2. \tag{10}$$

*If* $X_1, \ldots, X_n \in \mathbb{R}^d$ *are independent random variables and* $\mathbb{E}X_i = \boldsymbol{0}$ *for all* $i$, *then*

$$\mathbb{E}\left\|\sum_{i=1}^{n} X_i\right\|^2 = \sum_{i=1}^{n} \mathbb{E}\|X_i\|^2. \tag{11}$$

**Proposition 3.** *If* $X, Y \in \mathbb{R}^d$ *are two random variables (possibly dependent), then*

$$\mathbb{E}\|X + Y\|^2 \leq \|\mathbb{E}X + \mathbb{E}Y\|^2 + 2\mathbb{E}\|X - \mathbb{E}X\|^2 + 2\mathbb{E}\|Y - \mathbb{E}Y\|^2. \tag{12}$$

*Proof.*

$$\mathbb{E}\|X + Y\|^2 \overset{(8)}{=} \|\mathbb{E}X + \mathbb{E}Y\|^2 + \mathbb{E}\|X + Y - \mathbb{E}(X + Y)\|^2$$

$$\overset{(5)}{\leq} \|\mathbb{E}X + \mathbb{E}Y\|^2 + 2\mathbb{E}\|X - \mathbb{E}X\|^2 + 2\mathbb{E}\|Y - \mathbb{E}Y\|^2.$$

$\square$

## D  FedPAGE WITH $K = 1$

In this section, we discuss and analyze a special case (i.e., the local steps $K = 1$) of our FedPAGE algorithm. When we discuss the special case under the nonconvex and convex setting, we do not need all of the functions $\{f_i\}$ to be $L$-smooth. Instead, we only need the following average $L$-smoothness assumption, which is a weaker assumption compared with the smoothness Assumption 1.

**Assumption 3** (Average $L$-smoothness). *A function* $f : \mathbb{R}^d \to \mathbb{R}$ *is average $L$-smooth if there exists* $L \geq 0$ *such that for all* $x_1, x_2 \in \mathbb{R}^d$,

$$\mathbb{E}_i\|\nabla f_i(x_1) - \nabla f_i(x_2)\|^2 \leq L^2\|x_1 - x_2\|^2.$$

If the functions $\{f_i\}$ are average $L$-smooth (Assumption 3), then $f(x) = \frac{1}{N}\sum_{i=1}^{N} f_i(x)$ is also $L$-smooth, i.e., for all $x_1, x_2 \in \mathbb{R}^d$, $\|\nabla f(x_1) - \nabla f(x_2)\|_2 \leq L\|x_1 - x_2\|_2$.

### D.1 FedPAGE WITH LOCAL STEP $K = 1$

In this section, we review the convergence rate of PAGE in the nonconvex setting. The following theorem is directly derived from Theorem 1 in (Li et al., 2021; Li, 2021b).

**Theorem 3** (Theorem 1 in (Li et al., 2021)). *Suppose that Assumption 3 holds, i.e. $\{f_i\}$ are average $L$-smooth. If we choose the sampling probability $p_0 = 1$ and $p_r \equiv p = \frac{S}{N}$ for every $r \geq 1$, the global step size*

$$\eta_g = \frac{1}{L(1+\sqrt{\frac{1-p}{2pS}})},$$

*then FedPAGE with $K = 1$ (PAGE) will find a point $x$ such that $\mathbb{E}\|\nabla f(x)\|_2 \leq \epsilon$ with the number of communication rounds bounded by*

$$R = O\left(\frac{L(\sqrt{N}+S)}{S\epsilon^2}\right).$$

Li et al. (2021) also provide the tight lower bound (Theorem 2 of (Li et al., 2021)) indicating that the convergence result of PAGE (i.e., Theorem 3) is optimal in this nonconvex setting.

### D.2 FedPAGE WITH LOCAL STEP $K = 1$

Now we show the convergence result of FedPAGE with $K = 1$ (PAGE) in the convex setting. We assume that the functions $\{f_i\}$ are average $L$-smooth and function $f = \frac{1}{N}\sum_{i=1}^{n} f_i(x)$ is convex.

**Theorem 4** (Convergence of FedPAGE in convex setting when $K = 1$). *Suppose that $f$ is convex and Assumption 3 holds, i.e. $\{f_i\}$ are average $L$-smooth. If we choose the sampling probability $p_0 = 1$ and $p_r \equiv p = \frac{S}{N}$ for every $r \geq 1$, the number of local steps $K = 1$, the minibatch sizes $b_1 = b_2 = M$, the global step size*

$$\eta_g \leq \Theta\left(\frac{(S+N^{3/4})\frac{S}{N}}{L(S+\sqrt{N})}\right),$$

*then FedPAGE will find a point $x$ such that $\mathbb{E}f(x) - f^* \leq \epsilon$ with the number of communication rounds bounded by*

$$R = \begin{cases} O\left(\frac{N^{3/4}L}{S\epsilon}\right), & \text{if } S \leq \sqrt{N} \\ O\left(\frac{N^{1/4}L}{\epsilon}\right), & \text{if } \sqrt{N} < S \leq N^{3/4} \\ O\left(\frac{NL}{S\epsilon}\right), & \text{if } N^{3/4} < S \end{cases}.$$

To understand this result, we can set $S = 1$, i.e., in each round as long as the server does not communicate with all clients, it only selects one client to communicate. Then, the total communication cost of FedPAGE (here also the convergence result for PAGE) becomes $O\left(\frac{N^{3/4}}{\epsilon}\right)$. Recall that in the convex setting, SVRG/SAGA has convergence result $O\left(\frac{N}{\epsilon}\right)$. Thus FedPAGE/PAGE has much better convergence result compared with SVRG/SAGA in terms of the total number of clients $N$.

## E   MISSING PROOFS IN SECTION 4

In this section, we prove the convergence result of FedPAGE in the nonconvex setting (Theorem 1).

We use $\mathbb{E}_r$ to denote the expectation after $x^r$ is determined. Recall that we assume that $\{f_{i,j}\}$ are $L$-smooth, and formally, we have the following assumption

**Assumption 1** (*$L$-smoothness*). *All functions $f_{i,j} : \mathbb{R}^d \to \mathbb{R}$ for all $i \in [N], j \in [M]$ are $L$-smooth. That is, there exists $L \geq 0$ such that for all $x_1, x_2 \in \mathbb{R}^d$ and all $i \in [N], j \in [M]$,*

$$\|\nabla f_{i,j}(x_1) - \nabla f_{i,j}(x_2)\| \leq L\|x_1 - x_2\|.$$

**Lemma 1.** *Under Assumption 1, if we choose $b_3 = 1$ and the local step size $\eta_l \leq \frac{\sqrt{2}p}{24\sqrt{S}KL}$ in* FedPAGE*, we have for any $i, k, r$*

$$\frac{1}{K} \sum_{k=0}^{K-1} \mathbb{E}_r \|g_{i,k}^r - g_{i,0}^r\|^2$$

$$\leq 12K^2 L^2 \eta_l^2 \left( \frac{\sigma^2 \mathbb{I}\{b_2 < M\}}{b_2} + L^2 \|x^r - x^{r-1}\|^2 + \|g^{r-1} - \nabla f(x^{r-1})\|^2 + \|\nabla f(x^{r-1})\|^2 \right)$$

*Proof.* For any $i, k, r$, we have

$$\mathbb{E}_r \|g_{i,k}^r - g_{i,0}^r\|^2$$

$$= \mathbb{E}_r \|\nabla_{\mathcal{I}_3} f_i(y_{i,k}^r) - \nabla_{\mathcal{I}_3} f_i(y_{i,k-1}^r) + g_{i,k-1}^r - g_{i,0}^r\|^2$$

$$\overset{(5)}{=} \left(1 + \frac{1}{K-1}\right) \|g_{i,k-1}^r - g_{i,0}^r\|^2 + K \mathbb{E}_r \|\nabla_{\mathcal{I}_3} f_i(y_{i,k}^r) - \nabla_{\mathcal{I}_3} f_i(y_{i,k-1}^r)\|^2$$

$$\leq \left(1 + \frac{1}{K-1}\right) \mathbb{E}_r \|g_{i,k-1}^r - g_{i,0}^r\|^2 + K L^2 \mathbb{E}_r \|y_{i,k}^r - y_{i,k-1}^r\|^2 \qquad (13)$$

$$= \left(1 + \frac{1}{K-1}\right) \mathbb{E}_r \|g_{i,k-1}^r - g_{i,0}^r\|^2 + K L^2 \eta_l^2 \mathbb{E}_r \|g_{i,k-1}^r\|^2$$

$$= \left(1 + \frac{1}{K-1}\right) \mathbb{E}_r \|g_{i,k-1}^r - g_{i,0}^r\|^2 + K L^2 \eta_l^2 \mathbb{E}_r \|g_{i,k-1}^r - g_{i,0}^r + g_{i,0}^r\|^2$$

$$= \left(1 + \frac{1}{K-1}\right) \mathbb{E}_r \|g_{i,k-1}^r - g_{i,0}^r\|^2 + K L^2 \eta_l^2 \mathbb{E}_r \|g_{i,k-1}^r - g_{i,0}^r + \nabla_{\mathcal{I}_2} f_i(x^r) - \nabla_{\mathcal{I}_2} f_i(x^{r-1}) + g^{r-1}\|^2$$

$$\overset{(12)}{\leq} \left(1 + \frac{1}{K-1}\right) \mathbb{E}_r \|g_{i,k-1}^r - g_{i,0}^r\|^2 + K L^2 \eta_l^2 \mathbb{E}_r \|g_{i,k-1}^r - g_{i,0}^r + \nabla f_i(x^r) - \nabla f_i(x^{r-1}) + g^{r-1}\|^2$$

$$\quad + 4K L^2 \eta_l^2 \frac{\sigma^2 \mathbb{I}\{b_2 < M\}}{b_2}$$

$$= \left(1 + \frac{1}{K-1}\right) \mathbb{E}_r \|g_{i,k-1}^r - g_{i,0}^r\|^2 + 4K L^2 \eta_l^2 \frac{\sigma^2 \mathbb{I}\{b_2 < M\}}{b_2}$$

$$\quad + K L^2 \eta_l^2 \mathbb{E}_r \|g_{i,k-1}^r - g_{i,0}^r + \nabla f_i(x^r) - \nabla f_i(x^{r-1}) + g^{r-1} - \nabla f(x^{r-1}) + \nabla f(x^{r-1})\|^2$$

$$\overset{(6)}{\leq} \left(1 + \frac{1}{K-1}\right) \mathbb{E}_r \|g_{i,k-1}^r - g_{i,0}^r\|^2 + 4K L^2 \eta_l^2 \frac{\sigma^2 \mathbb{I}\{b_2 < M\}}{b_2} + 4K L^2 \eta_l^2 \mathbb{E}_r \|g_{i,k-1}^r - g_{i,0}^r\|^2$$

$$\quad + 4K L^2 \eta_l^2 \|\nabla f_i(x^r) - \nabla f_i(x^{r-1})\|^2 + 4K L^2 \eta_l^2 \|g^{r-1} - \nabla f(x^{r-1})\|^2 + 4K L^2 \eta_l^2 \|\nabla f(x^{r-1})\|^2$$

$$= \left(1 + \frac{1}{K-1} + 4K L^2 \eta_l^2\right) \mathbb{E}_r \|g_{i,k-1}^r - g_{i,0}^r\|^2 + 4K L^2 \eta_l^2 \frac{\sigma^2 \mathbb{I}\{b_2 < M\}}{b_2}$$

$$\quad + 4K L^2 \eta_l^2 \|\nabla f_i(x^r) - \nabla f_i(x^{r-1})\|^2 + 4K L^2 \eta_l^2 \|g^{r-1} - \nabla f(x^{r-1})\|^2 + 4K L^2 \eta_l^2 \|\nabla f(x^{r-1})\|^2$$

$$\leq 4K L^2 \eta_l^2 \left( \frac{\sigma^2 \mathbb{I}\{b_2 < M\}}{b_2} + \|\nabla f_i(x^r) - \nabla f_i(x^{r-1})\|^2 + \|g^{r-1} - \nabla f(x^{r-1})\|^2 + \|\nabla f(x^{r-1})\|^2 \right)$$

$$\quad \cdot \left( \sum_{k'=0}^{k-1} (1 + \frac{1}{K-1} + 4K L^2 \eta_l^2)^{k'} \right)$$

$$\leq 4K L^2 \eta_l^2 \left( \frac{\sigma^2 \mathbb{I}\{b_2 < M\}}{b_2} + \|\nabla f_i(x^r) - \nabla f_i(x^{r-1})\|^2 + \|g^{r-1} - \nabla f(x^{r-1})\|^2 + \|\nabla f(x^{r-1})\|^2 \right)$$

$$\quad \cdot \left( \sum_{k=0}^{K-1} (1 + \frac{1}{K-1} + 4K L^2 \eta_l^2)^{k} \right)$$

$$\leq 12K^2 L^2 \eta_l^2 \left( \frac{\sigma^2 \mathbb{I}\{b_2 < M\}}{b_2} + \|\nabla f_i(x^r) - \nabla f_i(x^{r-1})\|^2 + \|g^{r-1} - \nabla f(x^{r-1})\|^2 + \|\nabla f(x^{r-1})\|^2 \right)$$

$$(14)$$

$$\leq 12K^2L^2\eta_l^2\left(\frac{\sigma^2\mathbb{I}\{b_2 < M\}}{b_2} + L^2\|x^r - x^{r-1}\|^2 + \|g^{r-1} - \nabla f(x^{r-1})\|^2 + \|\nabla f(x^{r-1})\|^2\right).$$

In the derivation, (13) comes from the smoothness assumption (Assumption 1), (14) comes from the fact that if we choose $\eta_l \leq \frac{\sqrt{2}p}{24\sqrt{S}KL}$, then we have

$$\sum_{k=0}^{K-1}\left(1 + \frac{1}{K-1} + 4KL^2\eta_l^2\right)^k$$

$$\leq \frac{\left(1 + \frac{1}{K-1} + 4KL^2\eta_l^2\right)^K - 1}{\frac{1}{K-1} + 4KL^2\eta_l^2}$$

$$\leq (K-1)\left(\left(1 + \frac{1}{K-1} + 4KL^2\eta_l^2\right)^K - 1\right)$$

$$\leq (K-1)\left(\left(1 + \frac{1}{K-1} + \frac{1}{36K}\right)^K - 1\right)$$

$$\leq 3K,$$

for any $K \geq 2$. Then we take the average over $k$, we get

$$\frac{1}{K}\sum_{k=0}^{K-1}\mathbb{E}_r\|g_{i,k}^r - g_{i,0}^r\|^2$$

$$\leq 12K^2L^2\eta_l^2\left(\frac{\sigma^2\mathbb{I}\{b_2 < M\}}{b_2} + L^2\|x^r - x^{r-1}\|^2 + \|g^{r-1} - \nabla f(x^{r-1})\|^2 + \|\nabla f(x^{r-1})\|^2\right),$$

and we conclude the proof of this lemma. $\qquad\square$

**Lemma 2.** *Under Assumption 1, if we choose $b_3 = 1$ in* FedPAGE, *the local step size $\eta_l \leq \frac{\sqrt{2}p}{24\sqrt{S}KL}$, the batch sizes $b_1 = \min\{M, \frac{24\sigma^2}{pN\epsilon^2}\}$, $b_2 = \min\{M, \frac{24\sigma^2}{pS\epsilon^2}\}$, then we have*

$$\mathbb{E}_r\|g^r - \nabla f(x^r)\|^2 \leq (1-\frac{p}{3})\|g^{r-1} - \nabla f(x^{r-1})\|^2 + \frac{1-p/3}{S}L^2\mathbb{E}_r\|x^r - x^{r-1}\|^2 + \frac{p}{6S}\|\nabla f(x^{r-1})\|^2 + \frac{p\epsilon^2}{8}.$$

*Proof.*

$$\mathbb{E}_r\|g^r - \nabla f(x^r)\|^2$$

$$= (1-p)\mathbb{E}_r\left\|\frac{1}{K|S^r|}\sum_{i\in S^r}\sum_{k=1}^{K}g_{i,k-1}^r - \nabla f(x^r)\right\|^2 + p\left\|\frac{1}{N}\sum_{i\in[N]}\nabla_{\mathcal{I}_1}f_i(x^r) - \nabla f(x^r)\right\|^2$$

$$= (1-p)\mathbb{E}_r\left\|\frac{1}{K|S^r|}\sum_{i\in S^r}\sum_{k=1}^{K}g_{i,k-1}^r - \nabla f(x^r)\right\|^2 + \frac{p\sigma^2\mathbb{I}\{b_1 < M\}}{Nb_1}$$

$$= (1-p)\mathbb{E}_r\left\|\frac{1}{K|S^r|}\sum_{i\in S^r}\sum_{k=1}^{K}\left(g_{i,k-1}^r - g_{i,0}^r + g_{i,0}^r - \nabla f(x^r)\right)\right\|^2 + \frac{p\sigma^2\mathbb{I}\{b_1 < M\}}{Nb_1}$$

$$= (1-p)\mathbb{E}_r\left\|\frac{1}{K|S^r|}\sum_{i\in S^r}\sum_{k=1}^{K}\left(g_{i,k-1}^r - g_{i,0}^r + \nabla_{\mathcal{I}_2}f_i(x^r) - \nabla_{\mathcal{I}_2}f_i(x^{r-1}) + g^{r-1} - \nabla f(x^r)\right)\right\|^2$$
$$+ \frac{p\sigma^2\mathbb{I}\{b_1 < M\}}{Nb_1}$$

$$\overset{(8)}{=} (1-p)\left\|\mathbb{E}_r\frac{1}{K|S^r|}\sum_{i\in S^r}\sum_{k=1}^{K}\left(g_{i,k-1}^r - g_{i,0}^r + g^{r-1} - \nabla f(x^{r-1})\right)\right\|^2 + \frac{p\sigma^2\mathbb{I}\{b_1 < M\}}{Nb_1} \qquad (15)$$

$$+ (1-p)\mathbb{E}_r \left\| \frac{1}{K|S^r|} \sum_{i \in S^r} \sum_{k=1}^{K} \left(g_{i,k-1}^r - g_{i,0}^r\right) - \mathbb{E}_r \frac{1}{K|S^r|} \sum_{i \in S^r} \sum_{k=1}^{K} \left(g_{i,k-1}^r - g_{i,0}^r\right)\right.$$

$$\left. + \frac{1}{K|S^r|} \sum_{i \in S^r} \sum_{k=1}^{K} \left(\nabla_{\mathcal{I}_2} f_i(x^r) - \nabla f(x^r) + \nabla f(x^{r-1}) - \nabla_{\mathcal{I}_2} f_i(x^{r-1})\right)\right\|^2$$

$$\overset{(5)}{\leq} (1-p)\left(1+\frac{p}{2}\right) \|g^{r-1} - \nabla f(x^{r-1})\|^2 + \frac{p\sigma^2 \mathbb{I}\{b_1 < M\}}{Nb_1}$$

$$+ (1-p)\left(1+\frac{2}{p}\right) \left\| \mathbb{E}_r \frac{1}{K|S^r|} \sum_{i \in S^r} \sum_{k=1}^{K} \left(g_{i,k-1}^r - g_{i,0}^r\right)\right\|^2$$

$$+ (1-p)\left(1+\frac{p}{2}\right) \mathbb{E}_r \left\| \frac{1}{K|S^r|} \sum_{i \in S^r} \sum_{k=1}^{K} \left(\nabla_{\mathcal{I}_2} f_i(x^r) - \nabla f(x^r) + \nabla f(x^{r-1}) - \nabla_{\mathcal{I}_2} f_i(x^{r-1})\right)\right\|^2$$

$$+ (1-p)\left(1+\frac{2}{p}\right) \mathbb{E}_r \left\| \frac{1}{K|S^r|} \sum_{i \in S^r} \sum_{k=1}^{K} \left(g_{i,k-1}^r - g_{i,0}^r\right) - \mathbb{E}_r \frac{1}{K|S^r|} \sum_{i \in S^r} \sum_{k=1}^{K} \left(g_{i,k-1}^r - g_{i,0}^r\right)\right\|^2$$

$$= (1-p)\left(1+\frac{p}{2}\right) \|g^{r-1} - \nabla f(x^{r-1})\|^2 + \frac{p\sigma^2 \mathbb{I}\{b_1 < M\}}{Nb_1}$$

$$+ (1-p)\left(1+\frac{2}{p}\right) \left\| \mathbb{E}_r \frac{1}{K|S^r|} \sum_{i \in S^r} \sum_{k=1}^{K} \left(g_{i,k-1}^r - g_{i,0}^r\right)\right\|^2$$

$$+ (1-p)\left(1+\frac{p}{2}\right) \mathbb{E}_r \left\| \frac{1}{|S^r|} \sum_{i \in S^r} \left(\nabla_{\mathcal{I}_2} f_i(x^r) - \nabla f(x^r) + \nabla f(x^{r-1}) - \nabla_{\mathcal{I}_2} f_i(x^{r-1})\right)\right\|^2$$

$$+ (1-p)\left(1+\frac{2}{p}\right) \mathbb{E}_r \left\| \frac{1}{K|S^r|} \sum_{i \in S^r} \sum_{k=1}^{K} \left(g_{i,k-1}^r - g_{i,0}^r\right) - \mathbb{E}_r \frac{1}{K|S^r|} \sum_{i \in S^r} \sum_{k=1}^{K} \left(g_{i,k-1}^r - g_{i,0}^r\right)\right\|^2$$

$$\overset{(11)}{\leq} (1-p)\left(1+\frac{p}{2}\right) \|g^{r-1} - \nabla f(x^{r-1})\|^2 + \frac{p\sigma^2 \mathbb{I}\{b_1 < M\}}{Nb_1}$$

$$+ (1-p)\left(1+\frac{2}{p}\right) \left\| \mathbb{E}_r \frac{1}{K|S^r|} \sum_{i \in S^r} \sum_{k=1}^{K} \left(g_{i,k-1}^r - g_{i,0}^r\right)\right\|^2$$

$$+ \frac{1-p}{S}\left(1+\frac{p}{2}\right) \mathbb{E}_r \left\| \nabla_{\mathcal{I}_2} f_i(x^r) - \nabla f(x^r) + \nabla f(x^{r-1}) - \nabla_{\mathcal{I}_2} f_i(x^{r-1})\right\|^2 \qquad (16)$$

$$\frac{1-p}{S}\left(1+\frac{2}{p}\right) \mathbb{E}_r \left\| \frac{1}{K} \sum_{k=1}^{K} \left(g_{i,k-1}^r - g_{i,0}^r\right) - \mathbb{E}_r \frac{1}{K} \sum_{k=1}^{K} \left(g_{i,k-1}^r - g_{i,0}^r\right)\right\|^2$$

$$\overset{(10)}{\leq} (1-p)\left(1+\frac{p}{2}\right) \|g^{r-1} - \nabla f(x^{r-1})\|^2 + \frac{p\sigma^2 \mathbb{I}\{b_1 < M\}}{Nb_1}$$

$$+ (1-p)\left(1+\frac{2}{p}\right) \left\| \mathbb{E}_r \frac{1}{K|S^r|} \sum_{i \in S^r} \sum_{k=1}^{K} \left(g_{i,k-1}^r - g_{i,0}^r\right)\right\|^2$$

$$+ \frac{1-p}{S}\left(1+\frac{p}{2}\right) \mathbb{E}_r \left\| \left(\nabla f_i(x^r) - \nabla f(x^r) + \nabla f(x^{r-1}) - \nabla f_i(x^{r-1})\right)\right\|^2$$

$$+ \frac{1-p}{S}\left(1+\frac{p}{2}\right) \frac{4\sigma^2 \mathbb{I}\{b_2 < M\}}{b_2}$$

$$+ \frac{1-p}{S}\left(1+\frac{2}{p}\right) \mathbb{E}_r \left\| \frac{1}{K} \sum_{k=1}^{K} \left(g_{i,k-1}^r - g_{i,0}^r\right) - \mathbb{E}_r \frac{1}{K} \sum_{k=1}^{K} \left(g_{i,k-1}^r - g_{i,0}^r\right)\right\|^2$$

$$\overset{(9)}{\leq}(1-\frac{p}{2})\|g^{r-1}-\nabla f(x^{r-1})\|^2 + \frac{p\sigma^2\mathbb{I}\{b_1 < M\}}{Nb_1}$$

$$+ \frac{2}{p}\left\|\mathbb{E}_r\frac{1}{K|S^r|}\sum_{i\in S^r}\sum_{k=1}^K\left(g_{i,k-1}^r - g_{i,0}^r\right)\right\|^2$$

$$+ \frac{1-p/2}{S}\mathbb{E}_r\left\|\left(\nabla f_i(x^r) - \nabla f_i(x^{r-1})\right)\right\|^2 + \frac{1-p/2}{S}\frac{4\sigma^2\mathbb{I}\{b_2 < M\}}{b_2}$$

$$+ \frac{2}{pS}\mathbb{E}_r\left\|\frac{1}{K}\sum_{k=1}^K\left(g_{i,k-1}^r - g_{i,0}^r\right)\right\|^2. \tag{17}$$

In the previous derivations, (15) comes from the fact that we separate the mean and the variance of a random variable (Equation (8)). In (16), we define $z_t$ to be $\nabla_{\mathcal{I}_2}f_i(x^r) - \nabla f(x^r) + \nabla f(x^{r-1}) - \nabla_{\mathcal{I}_2}f_i(x^{r-1})$ and then apply (11). Here, $z_t$ are i.i.d. random variables.

Then we plug in Lemma 1, and by setting $\eta_l \leq \frac{\sqrt{2}p}{24\sqrt{S}KL}$, we have

$$\mathbb{E}_r\|g^r - \nabla f(x^r)\|^2$$

$$\overset{(17)}{\leq}\left(1-\frac{p}{2}\right)\|g^{r-1}-\nabla f(x^{r-1})\|^2 + \frac{p\sigma^2\mathbb{I}\{b_1 < M\}}{Nb_1}$$

$$+ \frac{2}{p}\left\|\mathbb{E}_r\frac{1}{K|S^r|}\sum_{i\in S^r}\sum_{k=1}^K\left(g_{i,k-1}^r - g_{i,0}^r\right)\right\|^2$$

$$+ \frac{1-p/2}{S}\mathbb{E}_r\left\|\left(\nabla f_i(x^r) - \nabla f_i(x^{r-1})\right)\right\|^2 + \frac{1-p/2}{S}\frac{4\sigma^2\mathbb{I}\{b_2 < M\}}{b_2}$$

$$+ \frac{2}{pS}\mathbb{E}_r\left\|\frac{1}{K}\sum_{k=1}^K\left(g_{i,k-1}^r - g_{i,0}^r\right)\right\|^2$$

$$\leq\left(1-\frac{p}{2}\right)\|g^{r-1}-\nabla f(x^{r-1})\|^2 + \frac{p\sigma^2\mathbb{I}\{b_1 < M\}}{Nb_1} \tag{18}$$

$$+ \frac{1-p/2}{S}L^2\mathbb{E}_r\|x^r - x^{r-1}\|^2 + \frac{1-p/2}{S}\frac{4\sigma^2\mathbb{I}\{b_2 < M\}}{b_2}$$

$$+ \frac{4}{p}\cdot 12K^2L^2\eta_l^2\left(\frac{\sigma^2\mathbb{I}\{b_2 < M\}}{b_2} + L^2\|x^r - x^{r-1}\|^2 + \|g^{r-1}-\nabla f(x^{r-1})\|^2 + \|\nabla f(x^{r-1})\|^2\right)$$

$$\overset{\text{Plug in }\eta_l}{\leq}(1-\frac{p}{2})\|g^{r-1}-\nabla f(x^{r-1})\|^2 + \frac{p\sigma^2\mathbb{I}\{b_1 < M\}}{Nb_1}$$

$$+ \frac{1-p/2}{S}L^2\mathbb{E}_r\|x^r - x^{r-1}\|^2 + \frac{1-p/2}{S}\frac{4\sigma^2\mathbb{I}\{b_2 < M\}}{b_2}$$

$$+ \frac{p}{6S}\frac{\sigma^2\mathbb{I}\{b_2 < M\}}{b_2} + \frac{pL^2}{6S}\|x^r - x^{r-1}\|^2 + \frac{p}{6S}\|g^{r-1}-\nabla f(x^{r-1})\|^2 + \frac{48K^2L^2\eta_l^2}{p}\|\nabla f(x^{r-1})\|^2$$

$$\leq(1-\frac{p}{3})\|g^{r-1}-\nabla f(x^{r-1})\|^2 + \frac{1-p/3}{S}L^2\mathbb{E}_r\|x^r - x^{r-1}\|^2 + \frac{48K^2L^2\eta_l^2}{p}\|\nabla f(x^{r-1})\|^2$$

$$+ \frac{p}{6S}\frac{\sigma^2\mathbb{I}\{b_2 < M\}}{b_2} + \frac{1-p/2}{S}\frac{4\sigma^2\mathbb{I}\{b_2 < M\}}{b_2} + \frac{p\sigma^2\mathbb{I}\{b_1 < M\}}{Nb_1},$$

where in (18), we apply Lemma 1 and Eq. (4). Plugging in the batch sizes $b_1 = \min\{M, \frac{24\sigma^2}{pN\epsilon^2}\}, b_2 = \min\{M, \frac{48\sigma^2}{pS\epsilon^2}\}$ and recall $\eta_l \leq \frac{\sqrt{2}p}{24\sqrt{S}KL}$, we get

$$\mathbb{E}_r\|g^r - \nabla f(x^r)\|^2$$

$$\leq\left(1-\frac{p}{3}\right)\|g^{r-1}-\nabla f(x^{r-1})\|^2 + \frac{1-p/3}{S}L^2\mathbb{E}_r\|x^r - x^{r-1}\|^2 + \frac{48K^2L^2\eta_l^2}{p}\|\nabla f(x^{r-1})\|^2 + \frac{p\epsilon^2}{8} \tag{19}$$

$$\leq (1 - \frac{p}{3})\|g^{r-1} - \nabla f(x^{r-1})\|^2 + \frac{1 - p/3}{S}L^2\mathbb{E}_r\|x^r - x^{r-1}\|^2 + \frac{p}{6S}\|\nabla f(x^{r-1})\|^2 + \frac{p\epsilon^2}{8}.$$

$$\square$$

Then, combining with the following descent lemma, we can prove Theorem 1.

**Lemma 3** (Lemma 2 in PAGE Li et al. (2021)). *Suppose that $f$ is $L$-smooth and let $x^{t+1} := x^t - \eta g^t$. Then we have*

$$f(x^{t+1}) \leq f(x^t) - \frac{\eta}{2}\|\nabla f(x^t)\|^2 - \left(\frac{1}{2\eta} - \frac{L}{2}\right)\|x^{t+1} - x^t\|^2 + \frac{\eta}{2}\|g^t - \nabla f(x^t)\|^2.$$

**Theorem 1** (Convergence of FedPAGE in nonconvex setting). *Under Assumption 1 (and Assumption 2), if we choose the sampling probability $p_r \equiv p = \frac{S}{N}$ for every $r \geq 1$ and $p_0 = 1$, the minibatch sizes $b_1 = \min\{M, \frac{24\sigma^2}{S\epsilon^2}\}, b_2 = \min\{M, \frac{48\sigma^2}{pS\epsilon^2}\}$, and the global and local step sizes*

$$\eta_g \leq \frac{1}{L\left(1 + \sqrt{\frac{3(1-p/3)}{2pS}}\right)}, \qquad \eta_l \leq \frac{\sqrt{2p}}{24\sqrt{S}KL},$$

*then FedPAGE will find a point $x$ such that $\mathbb{E}\|\nabla f(x)\|_2 \leq \epsilon$ within the following number of communication rounds:*

$$R = O\left(\frac{L(\sqrt{N} + S)}{S\epsilon^2}\right).$$

*Proof.* When $\eta_l < \frac{\sqrt{2p}}{24\sqrt{S}KL}$, Lemma 2 holds. If we choose the batch sizes $b_1 = \min\{M, \frac{24\sigma^2}{pN\epsilon^2}\}, b_2 = \min\{M, \frac{48\sigma^2}{pS\epsilon^2}\}$, we have

$$\mathbb{E}\left[f(x^r) - f^* + \frac{3\eta_g}{2p}\|g^r - \nabla f(x^r)\|^2\right]$$

$$\leq \mathbb{E}\left[f(x^{r-1}) - f^* - \frac{\eta_g}{2}\|\nabla f(x^{r-1})\|^2 - \left(\frac{1}{2\eta_g} - \frac{L}{2}\right)\|x^r - x^{r-1}\|^2 + \frac{\eta_g}{2}\|g^{r-1} - \nabla f(x^{r-1})\|^2\right]$$

$$\tag{20}$$

$$+ \frac{3\eta_g}{2p}\mathbb{E}\left[\left(1 - \frac{p}{3}\right)\|\nabla f(x^{r-1}) - g^{r-1}\|^2 + \frac{1}{S}\left(1 - \frac{p}{3}\right)L^2\mathbb{E}_r\|x^r - x^{r-1}\|^2 + \frac{p}{6S}\mathbb{E}_r\|\nabla f(x^{r-1})\|^2 + \frac{p\epsilon^2}{8}\right]$$

$$\leq \mathbb{E}[f(x^{r-1}) - f^* - \frac{\eta_g}{4}\|\nabla f(x^{r-1})\|^2 - \left(\frac{1}{2\eta_g} - \frac{L}{2} - \frac{3\eta_g}{2p}\frac{1}{S}\left(1 - \frac{p}{3}\right)L^2\right)\|x^r - x^{r-1}\|^2$$

$$+ \frac{3\eta_g}{2p}\|g^{r-1} - \nabla f(x^{r-1})\|^2,$$

$$\tag{21}$$

where in (20) we plug in Lemma 2 and Lemma 3, and in (21) we rearrange the terms.

Choosing $\eta_g = \frac{1}{L\left(1 + \sqrt{\frac{3(1-\frac{p}{3})}{2pS}}\right)}$ and $p = \frac{S}{N}$, the coefficient of $\|x^r - x^{r-1}\|^2$ is greater than zero, and we can throw that term away (since $\|x^r - x^{r-1}\| \geq 0$ and the sign is minus). Then we have

$$\mathbb{E}\left[f(x^r) - f^* + \frac{3\eta_g}{2p}\|g^r - \nabla f(x^r)\|^2\right]$$

$$\leq \mathbb{E}\left[f(x^{r-1}) - f^* + \frac{3\eta_g}{2p}\|g^{r-1} - \nabla f(x^{r-1})\|^2\right] + \frac{3\eta_g\epsilon^2}{16} - \frac{\eta_g}{4}\mathbb{E}\|\nabla f(x^{r-1})\|^2.$$

We also know that in the first round,

$$\mathbb{E}\left\|\frac{1}{N}\sum_{i=1}^{N}\tilde{\nabla}_{b_1}f_i(x^0) - \nabla f(x^0)\right\|^2 = \frac{\sigma^2}{Nb_1} \leq \frac{p\epsilon^2}{24},$$

and we have

$$\mathbb{E}\left[f(x^r) - f^* + \frac{3\eta_g}{2p}\|g^r - \nabla f(x^r)\|^2\right] \le \mathbb{E}\left[f(x^0) - f^* + \frac{3\eta_g}{2p}\frac{p\epsilon^2}{24}\right] + \frac{3r\eta_g\epsilon^2}{16} - \frac{\eta_g}{4}\sum_{i=0}^{r}\mathbb{E}\|\nabla f(x^i)\|^2,$$

which leads to

$$\frac{\eta_g}{4}\sum_{i=0}^{r}\mathbb{E}\|\nabla f(x^i)\|^2 \le \mathbb{E}\left[f(x^0) - f^* + \frac{3\eta_g}{2p}\frac{p\epsilon^2}{24}\right] + \frac{3r\eta_g\epsilon^2}{16},$$

where we use the fact that $\|\cdot\|^2 \ge 0$ and $f(x) - f^* \ge 0$.

So in $O(1/(\eta_g\epsilon^2))$ number of rounds, FedPAGE can find a point $x$ such that $\mathbb{E}\|\nabla f(x)\|^2 \le \epsilon^2$, which leads to a point $x$ such that $\mathbb{E}\|\nabla f(x)\| \le \epsilon$. Then since

$$\frac{1}{\eta_g} = L\left(1 + \sqrt{\frac{3(1-p/3)}{2pS}}\right) = O\left(L\left(1 + \frac{\sqrt{N}}{S}\right)\right) = O\left(\frac{\sqrt{N}+S}{S}\right),$$

we know that FedPAGE can find a point $x$ such that $\mathbb{E}\|\nabla f(x)\| \le \epsilon$ in $O\left(\frac{L(\sqrt{N}+S)}{S\epsilon^2}\right)$ number of communication rounds. □

## F  MISSING PROOF IN SECTION 5

In this section, we show the convergence result of FedPAGE in the convex setting. We first show the result when the number of local steps is 1 ($K = 1$), where FedPAGE reduces to PAGE algorithm (Section F.1). Then, we show the result of FedPAGE in the convex setting with general number of local steps.

### F.1  PROOF OF THEOREM 4

Similar to the notations in the proof for the nonconvex setting, we use $\mathcal{F}_t$ to denote the filtration when we determine the "gradient" $g^{t-1}$ but not $g^t$, i.e. $x^t$ is determined but $x^{t+1}$ is not determined. We use $\mathbb{E}_j[\cdot]$ to denote $\mathbb{E}[\cdot|\mathcal{F}_j]$.

Recall that in this section, we assume the objective function $f$ is convex and all the functions $\{f_i\}$ are averaged $L$-smooth.

**Assumption 3** (Average $L$-smoothness). *A function $f : \mathbb{R}^d \to \mathbb{R}$ is average $L$-smooth if there exists $L \ge 0$ such that for all $x_1, x_2 \in \mathbb{R}^d$,*

$$\mathbb{E}_i\|\nabla f_i(x_1) - \nabla f_i(x_2)\|^2 \le L^2\|x_1 - x_2\|^2.$$

The main difficult to prove Theorem 4 is that FedPAGE uses biased gradient estimator, i.e.

$$\mathbb{E}g^r \ne \mathbb{E}\nabla f(x^r),$$

for most of the rounds $r$. During the derivation, we will encounter the following inner product term

$$\mathbb{E}\langle\nabla f(x^{r-1}) - g^{r-1}, x^r - x^*\rangle.$$

If the gradient estimator is unbiased, the above inner product is zero and we don't have to worry about this term. But when the gradient estimator is biased, we need to bound this term.

However, since the server using FedPAGE will communicate with all of the clients with probability $p_r$ in round $r$ to get the full gradient $\nabla f(x^r)$, the following property holds.

**Lemma 4.** *When the number of local steps is 1 ($K = 1$) and we choose the probability $p_r = p$ for all $r$, FedPAGE satisfies for any $r \ge 1$,*

$$\mathbb{E}_r[g^r - \nabla f(x^r)] = (1-p)(g^{r-1} - \nabla f(x^{r-1})).$$

*Proof.* If in round $r$, the server does not communicate with all the client and only communicate with a subset of clients $S^r$, then from the definition of FedPAGE, $\Delta y_i^r = -\eta_l g_{i,0}^r$ and we can get

$$
g^r = -\frac{1}{K\eta_l |S^r|} \sum_{i \in S^r} \Delta y_i^t = \frac{1}{|S^r|} \sum_{i \in S^r} g_{i,0}^r.
$$

We use $\mathbb{E}_\mathcal{I}$ to denote the expectation over the minibatch $\mathcal{I}_2$ to estimate the local full gradient. Then we have

$$
\begin{aligned}
\mathbb{E}_r[g^r - \nabla f(x^r)] =& (1-p)\mathbb{E}_r \left[ \frac{1}{|S^r|} \sum_{i \in S^r} g_{i,0}^r - \nabla f(x^r) \right] + p\mathbb{E}_r \left[ \frac{1}{N} \sum_{i \in [N]} \tilde{\nabla}_{b_1} f_i(x^r) - \nabla f(x^r) \right] \\
=& (1-p)\frac{1}{|S^r|} \mathbb{E}_r \sum_{i \in S^r} \mathbb{E}_\mathcal{I} \left[ (g_{i,0}^r - \nabla f(x^r)) \right] \\
=& (1-p)\frac{1}{|S^r|} \mathbb{E}_r \sum_{i \in S^r} \mathbb{E}_\mathcal{I} \left[ \nabla_{\mathcal{I}_2} f_i(x^r) - \nabla_{\mathcal{I}_2} f_i(x^{r-1}) + g^{r-1} - \nabla f(x^r) \right] \\
=& (1-p)\frac{1}{|S^r|} \mathbb{E}_r \sum_{i \in S^r} \mathbb{E}_\mathcal{I} \left[ \nabla_{\mathcal{I}_2} f_i(x^r) - \nabla_{\mathcal{I}_2} f_i(x^{r-1}) + g^{r-1} - \nabla f(x^r) \right] \\
=& (1-p)\frac{1}{|S^r|} \mathbb{E}_r \sum_{i \in S^r} \left[ \nabla f_i(x^r) - \nabla f_i(x^{r-1}) + g^{r-1} - \nabla f(x^r) \right] \\
=& (1-p)(g^{r-1} - \nabla f(x^{r-1})),
\end{aligned}
$$

where we use the fact that $S^r$ is uniformly chosen from $[N]$ and $\nabla_{\mathcal{I}_2} f(x)$ is a gradient estimator of $\nabla f(x)$. $\qquad\square$

**Lemma 5** (Lemma 3 of (Li et al., 2021)))**.** *When the number of local steps is 1 and we choose $p_r = p$ for all $r$, FedPAGE satisfies for any $r \geq 1$,*

$$
\mathbb{E}_r \| g^r - \nabla f(x^r) \|_2^2 = (1-p)\| g^{r-1} - \nabla f(x^{r-1}) \|_2^2 + \frac{1-p}{S} \mathbb{E}_r \| \nabla f_i(x^r) - \nabla f_i(x^{r-1}) \|_2^2.
$$

Then, we can control the inner product term using the following lemma.

**Lemma 6.** *For any $t \geq 2$ and any $c > 0$, we have*

$$
\sum_{r=1}^t \mathbb{E}\langle \nabla f(x^{r-1}) - g^{r-1}, x^r - x^* \rangle \leq \frac{1}{2p} \mathbb{E} \sum_{r=1}^t \left( c\|\nabla f(x^{r-1}) - g^{r-1}\|_2^2 + \frac{1}{c}\|x^r - x^{r-1}\|_2^2 \right).
$$

*Proof.*

$$
\begin{aligned}
& \mathbb{E}\langle \nabla f(x^{r-1}) - g^{r-1}, x^r - x^* \rangle \\
=& \mathbb{E}\langle \nabla f(x^{r-1}) - g^{r-1}, x^r - x^{r-1} \rangle + \mathbb{E}\mathbb{E}_{r-1}\langle \nabla f(x^{r-1}) - g^{r-1}, x^{r-1} - x^* \rangle \\
\leq& \frac{c}{2} \mathbb{E}\|f(x^{r-1}) - g^{r-1}\|_2^2 + \frac{1}{2c}\mathbb{E}\|x^r - x^{r-1}\|_2^2 + (1-p)\mathbb{E}\langle \nabla f(x^{r-2}) - g^{r-2}, x^{r-1} - x^* \rangle,
\end{aligned}
$$

where the last inequality comes from Young's inequality and Lemma 4. For $r = 1$, we know that $\mathbb{E}_{r-1}\langle \nabla f(x^{r-1}) - g^{r-1}, x^{r-1} - x^* \rangle = 0$. Unrolling the inequality recursively, we have

$$
\begin{aligned}
& \mathbb{E}\langle \nabla f(x^{r-1}) - g^{r-1}, x^r - x^* \rangle \\
\leq& \frac{c}{2} \mathbb{E}\|f(x^{r-1}) - g^{r-1}\|_2^2 + \frac{1}{2c}\mathbb{E}\|x^r - x^{r-1}\|_2^2 + (1-p)\mathbb{E}\langle \nabla f(x^{r-2}) - g^{r-2}, x^{r-1} - x^* \rangle \\
\leq& \frac{c}{2} \mathbb{E}\|f(x^{r-1}) - g^{r-1}\|_2^2 + \frac{1}{2c}\mathbb{E}\|x^r - x^{r-1}\|_2^2 \\
& + \frac{1-p}{2}\mathbb{E}\left( c\|f(x^{r-2}) - g^{r-2}\|_2^2 + \frac{1}{c}\|x^{r-1} - x^{r-2}\|_2^2 \right) \\
& + (1-p)^2 \mathbb{E}\langle \nabla f(x^{r-3}) - g^{r-3}, x^{r-2} - x^* \rangle
\end{aligned}
$$

$$\leq \sum_{r'=1}^{r} (1-p)^{r-r'} \frac{c}{2} \mathbb{E} \|f(x^{r'-1}) - g^{r'-1}\|_2^2 + \sum_{r'=1}^{r} (1-p)^{r-r'} \frac{1}{2c} \mathbb{E} \|x^{r'} - x^{r'-1}\|_2^2.$$

Then we sum up the inequalities from $r = 1$ to $t$, we have

$$\sum_{r=1}^{t} \mathbb{E}\langle \nabla f(x^{r-1}) - g^{r-1}, x^r - x^* \rangle \leq \frac{1}{2p} \mathbb{E} \sum_{r=1}^{t} \left( c \|\nabla f(x^{r-1}) - g^{r-1}\|_2^2 + \frac{1}{c} \|x^r - x^{r-1}\|_2^2 \right).$$

$\square$

Given these lemmas, we can now prove Theorem 2. We first prove 2 lemmas related to the function decent of each step, and then show the proof of Theorem 2.

**Lemma 7.** *For any $r \geq 0$ and any $\lambda > 0$, we have*

$$0 \leq -\eta_g \mathbb{E}_r[f(x^{r+1}) - f(x^*)] + \frac{\eta_g}{2L\lambda} \mathbb{E}_r \|g^r - \nabla f(x^r)\|^2 - \frac{1}{2} \mathbb{E}_r \|x^{r+1} - x^*\|^2 + \frac{1}{2} \|x^r - x^*\|^2$$

$$\left( \frac{\eta_g L(\lambda+1)}{2} - \frac{1}{2} \right) \mathbb{E}_r \|x^{r+1} - x^r\|^2 + \eta_g(1-p)\langle \nabla f(x^{r-1}) - g^{r-1}, x^r - x^* \rangle.$$

*Here, we define $g^{-1} = \nabla f(x^{-1}) = 0$.*

*Proof.* For any $r \geq 0$, we have

$$\eta_g(f(x^r) - f(x^*))$$
$$\leq \eta_g \langle \nabla f(x^r), x^r - x^* \rangle$$
$$= \eta_g \langle \nabla f(x^r) - (1-p)(\nabla f(x^{r-1}) - g^{r-1}), x^r - x^* \rangle + \eta_g \langle (1-p)(\nabla f(x^{r-1}) - g^{r-1}), x^r - x^* \rangle$$
$$= \eta_g \mathbb{E}_r \langle g^r, x^r - x^* \rangle + \eta_g \langle (1-p)(\nabla f(x^{r-1}) - g^{r-1}), x^r - x^* \rangle$$
$$= \eta_g \mathbb{E}_r \langle g^r, x^r - x^{r+1} \rangle + \eta_g \mathbb{E}_r \langle g^r, x^{r+1} - x^* \rangle + \eta_g \langle (1-p)(\nabla f(x^{r-1}) - g^{r-1}), x^r - x^* \rangle$$
$$\leq \eta_g \mathbb{E}_r \langle g^r, x^r - x^{r+1} \rangle - \frac{1}{2} \mathbb{E}_r \|x^{r+1} - x^r\|^2 + \frac{1}{2} \mathbb{E}_r \|x^r - x^*\|^2 - \frac{1}{2} \mathbb{E}_r \|x^{r+1} - x^*\|^2$$
$$\quad + \eta_g \langle (1-p)(\nabla f(x^{r-1}) - g^{r-1}), x^r - x^* \rangle.$$

We also have

$$\eta_g \mathbb{E}_r \langle g^r, x^r - x^{r+1} \rangle$$
$$= \eta_g \mathbb{E}_r \langle g^r - \nabla f(x^r), x^r - x^{r+1} \rangle + \eta_g \mathbb{E}_r \langle \nabla f(x^r), x^r - x^{r+1} \rangle$$
$$\leq \frac{\eta_g}{2\lambda L} \mathbb{E}_r \|g^r - \nabla f(x^r)\|^2 + \frac{\eta_g \lambda L}{2} \mathbb{E}_r \|x^r - x^{r+1}\|^2 + \eta_g(f(x^r) - f(x^{r+1})) + \frac{\eta_g L}{2} \|x^{r+1} - x^r\|^2.$$

Summing up the 2 inequalities we conclude the proof. $\square$

**Lemma 8.** *For any $r \geq 0$ and any $\lambda > 0$, we have*

$$0 \leq \eta_g(f(x^r) - f(x^{r+1})) + \frac{\eta_g}{2L\lambda} \|g^r - f(x^r)\|^2 + \left( \frac{\eta_g L(\lambda+1)}{2} - 1 \right) \|x^r - x^{r+1}\|^2.$$

*Proof.*

$$0 = \eta_g \langle g^r, x^r - x^{r+1} \rangle + \eta_g \langle g^r, x^{r+1} - x^r \rangle$$
$$= \eta_g \langle \nabla f(x^r), x^r - x^{r+1} \rangle + \eta_g \langle g^r - \nabla f(x^r), x^r - x^{r+1} \rangle - \|x^{r+1} - x^r\|^2$$
$$\leq \eta_g(f(x^r) - f(x^{r+1})) + \eta_g \langle g^r - \nabla f(x^r), x^r - x^{r+1} \rangle + \left( \frac{\eta_g L}{2} - 1 \right) \|x^{r+1} - x^r\|^2$$
$$\leq \eta_g(f(x^r) - f(x^{r+1})) + \frac{\eta_g}{2\lambda L} \|g^r - \nabla f(x^r)\|^2 + \left( \frac{\eta_g L(\lambda+1)}{2} - 1 \right) \|x^{r+1} - x^r\|^2.$$

$\square$

**Theorem 4** (Convergence of FedPAGE in convex setting when $K = 1$). *Suppose that $f$ is convex and Assumption 3 holds, i.e. $\{f_i\}$ are average $L$-smooth. If we choose the sampling probability $p_0 = 1$ and $p_r \equiv p = \frac{S}{N}$ for every $r \geq 1$, the number of local steps $K = 1$, the minibatch sizes $b_1 = b_2 = M$, the global step size*

$$\eta_g \leq \Theta\left(\frac{(S + N^{3/4})\frac{S}{N}}{L(S + \sqrt{N})}\right),$$

*then FedPAGE will find a point $x$ such that $\mathbb{E}f(x) - f^* \leq \epsilon$ with the number of communication rounds bounded by*

$$R = \begin{cases} O\left(\dfrac{N^{3/4}L}{S\epsilon}\right), & \text{if } S \leq \sqrt{N} \\[2mm] O\left(\dfrac{N^{1/4}L}{\epsilon}\right), & \text{if } \sqrt{N} < S \leq N^{3/4} \\[2mm] O\left(\dfrac{NL}{S\epsilon}\right), & \text{if } N^{3/4} < S \end{cases}.$$

*Proof of Theorem 4.* From Lemma 7 and Lemma 8, for any $\delta > 0$, we have

$$\eta_g \mathbb{E}_r[f(x^{r+1}) - f(x^*) + \delta(f(x^r) - f(x^{r+1}))]$$

$$\leq \frac{\eta_g(1+\delta)}{2L\lambda}\mathbb{E}_r\|g^r - \nabla f(x^r)\|^2 - \frac{1}{2}\mathbb{E}_r\|x^{r+1} - x^*\|^2 + \frac{1}{2}\|x^r - x^*\|^2$$

$$+ \left(\frac{\eta_g L(\lambda+1)(1+\delta)}{2} - \frac{1+2\delta}{2}\right)\mathbb{E}_r\|x^{r+1} - x^r\|^2 + \eta_g(1-p)\langle\nabla f(x^{r-1}) - g^{r-1}, x^r - x^*\rangle.$$

Summing up the inequalities from $r = 0$ to $T - 1$ and taking the expectation, we have

$$\sum_{r=0}^{T-1} \eta_g \mathbb{E}[f(x^{r+1}) - f(x^*)] + \eta_g \delta \mathbb{E}[f(x^T) - f(x^0)]$$

$$\leq -\frac{1}{2}\mathbb{E}\|x^T - x^*\|^2 + \frac{1}{2}\|x^0 - x^*\|^2 + \sum_{r=0}^{T-1}\frac{\eta_g(1+\delta)}{2L\lambda}\mathbb{E}\|g^r - \nabla f(x^r)\|^2$$

$$+ \sum_{r=0}^{T-1}\left((1+\delta)\frac{L(\lambda+1)\eta_g - 1}{2}\mathbb{E}\|x^{r+1} - x^r\|^2 + \eta_g(1-p)\mathbb{E}\langle\nabla f(x^{r-1}) - g^{r-1}, x^r - x^*\rangle\right)$$

$$\leq -\frac{1}{2}\mathbb{E}\|x^T - x^*\|^2 + \frac{1}{2}\|x^0 - x^*\|^2 + \sum_{r=0}^{T-1}\left(\frac{\eta_g(1+\delta)}{2L\lambda} + \frac{\eta_g(1-p)c}{2p}\right)\mathbb{E}\|g^r - \nabla f(x^r)\|^2$$

$$+ \sum_{r=0}^{T-1}(1+\delta)\left(\frac{L\eta_g(\lambda+1) - 1}{2} + \frac{\eta_g(1-p)}{2cp(1+\delta)}\right)\mathbb{E}\|x^{r+1} - x^r\|^2,$$

where we apply Lemma 6 to bound the inner product term. Then using Lemma 5 and Assumption 3, we can get the following result.

$$\sum_{r=0}^{T-1}\mathbb{E}\|g^r - \nabla f(x^r)\|^2 \leq \sum_{r=1}^{T-1}\frac{1-p}{pS}\mathbb{E}\mathbb{E}_r\|\nabla f_i(x^r) - \nabla f_i(x^{r-1})\|^2 \leq \sum_{r=1}^{T-1}\frac{(1-p)L^2}{pS}\mathbb{E}\|x^r - x^{r-1}\|^2.$$

Plugging into the previous inequality, we have

$$\sum_{r=0}^{T-1} \eta_g \mathbb{E}[f(x^{r+1}) - f(x^*)] + \eta_g \delta \mathbb{E}[f(x^T) - f(x^0)]$$

$$\leq -\frac{1}{2}\mathbb{E}\|x^T - x^*\|^2 + \frac{1}{2}\|x^0 - x^*\|^2 + w\sum_{r=0}^{T-1}\mathbb{E}\|x^{r+1} - x^r\|^2,$$

where

$$w = (1+\delta)\left(\frac{L\eta_g(\lambda+1)-1}{2} + \frac{\eta_g(1-p)}{2cp(1+\delta)}\right) + \left(\frac{\eta_g(1+\delta)}{2L\lambda} + \frac{\eta_g(1-p)c}{2p}\right)\frac{(1-p)L^2}{pS}.$$

By choosing $\lambda = \sqrt{1/(Sp)}, c = \sqrt{Sp/L^2}$, we have

$$w = \frac{(1+\delta)L\eta_g(\sqrt{1/(Sp)}+1)}{2} - \frac{1+\delta}{2} + \frac{\eta_g(1-p)}{2p\sqrt{pS/L^2}} + \left(\frac{\eta_g(1+\delta)}{2L\sqrt{1/(Sp)}} + \frac{\eta_g(1-p)\sqrt{Sp/L^2}}{2p}\right)\frac{(1-p)L^2}{pS}$$

$$= (1+\delta)\left(\frac{L\eta_g(\sqrt{1/(Sp)}+1)}{2} - \frac{1}{2} + \frac{\eta_g(1-p)L}{2p^{3/2}\sqrt{S}(1+\delta)} + \left(\frac{\eta_g\sqrt{Sp}}{2L} + \frac{\eta_g(1-p)\sqrt{Sp}}{2pL(1+\delta)}\right)\frac{(1-p)L^2}{pS}\right)$$

$$= (1+\delta)\left(\frac{L\eta_g(\sqrt{1/(Sp)}+1)}{2} - \frac{1}{2} + \frac{\eta_g(1-p)L}{2p^{3/2}\sqrt{S}(1+\delta)} + \left(\frac{\eta_g}{2} + \frac{\eta_g(1-p)}{2p(1+\delta)}\right)\frac{(1-p)L}{\sqrt{pS}}\right)$$

$$= (1+\delta)\left(\underbrace{\frac{L\eta_g(\sqrt{1/(Sp)}+1)}{2}}_{\mathcal{A}} + \underbrace{\frac{\eta_g(1-p)L}{2p^{3/2}\sqrt{S}(1+\delta)}}_{\mathcal{B}} + \underbrace{\frac{\eta_g}{2}\frac{(1-p)L}{\sqrt{pS}}}_{\mathcal{C}} + \underbrace{\frac{\eta_g(1-p)}{2p(1+\delta)}\frac{(1-p)L}{\sqrt{pS}}}_{\mathcal{D}} - \frac{1}{2}\right).$$

If $\eta_g \le O\left(\frac{(1+\delta)p}{L(1+\sqrt{1/(Sp)})}\right)$ and $\delta \le 1/p$, we have

$$\mathcal{A} = \frac{L\eta_g(\sqrt{1/(Sp)}+1)}{2} = O\left(L(\sqrt{1/(Sp)}+1)\frac{(1/p)p}{L(\sqrt{1/(Sp)}+1)}\right) = O(1),$$

$$\mathcal{B} = \frac{\eta_g(1-p)L}{2p^{3/2}\sqrt{S}(1+\delta)} = O\left(\frac{(1-p)L}{2p^{3/2}\sqrt{S}}\frac{p}{L(1+\sqrt{1/(Sp)})}\right) = O(1),$$

$$\mathcal{C} = \frac{\eta_g}{2}\frac{(1-p)L}{\sqrt{pS}} = O\left(\frac{(1-p)L}{\sqrt{pS}}\frac{(1/p)p}{L(\sqrt{1/(Sp)}+1)}\right) = O(1),$$

$$\mathcal{D} = \frac{(1-p)}{2p}\frac{(1-p)L}{\sqrt{pS}}\frac{p}{L(1+\sqrt{1/(Sp)})} = O(1).$$

In this way, we can choose $\eta_g$ with a small constant such that $w$ is non-positive, and we can throw that term. In this way,

$$\sum_{r=0}^{T-1}\mathbb{E}[f(x^{r+1}) - f(x^*)] \le \frac{1}{2\eta_g}\|x^0 - x^*\|^2 + \delta[f(x^0) - f(x^*)] \le \left(\frac{1}{2\eta_g} + \frac{L\delta}{2}\right)\|x^0 - x^*\|^2.$$

Then we set $\delta = 1/(S^{1/4}p^{3/4})$ and $p = S/N$. We first verify that $\delta \le 1/p$. We have

$$\delta = \frac{1}{S^{1/4}}\frac{N^{3/4}}{S^{3/4}} = \frac{N^{3/4}}{S} \le \frac{N}{S} = \frac{1}{p}.$$

Then we choose

$$\eta_g = \Theta\left(\frac{(1+\frac{N^{3/4}}{S})\frac{S}{N}}{L(1+\frac{\sqrt{N}}{S})}\right) = \Theta\left(\frac{(S+N^{3/4})\frac{S}{N}}{L(S+\sqrt{N})}\right) = \begin{cases} \Theta\left(\frac{S}{N^{3/4}L}\right), \text{ if } S \le \sqrt{N}, \\ \Theta\left(\frac{1}{N^{1/4}L}\right), \text{ if } \sqrt{N} < S \le N^{3/4}, \\ \Theta\left(\frac{S}{NL}\right), \text{ if } N^{3/4} < S \end{cases}$$

Then, the number of communication round is bounded by

$$R = \begin{cases} O\left(\frac{N^{3/4}L}{S\epsilon}\right), \text{ if } S \le \sqrt{N}, \\ O\left(N^{1/4}L/\epsilon\right), \text{ if } \sqrt{N} < S \le N^{3/4}, \\ O\left(\frac{NL}{S\epsilon}\right), \text{ if } N^{3/4} < S \end{cases}$$

$\qquad\square$

### F.2 Proof of Theorem 2

The proof idea of Theorem 2 is similar to that of Theorem 4. The difference between these two proof comes from the fact that in the convex setting with general local steps, the local steps between the communication rounds introduce some local error and we need to take the error into account.

In this section, we assume that all the functions $\{f_{i,j}\}$ are $L$-smooth.

**Assumption 1** ($L$-smoothness). *All functions $f_{i,j} : \mathbb{R}^d \to \mathbb{R}$ for all $i \in [N], j \in [M]$ are L-smooth. That is, there exists $L \geq 0$ such that for all $x_1, x_2 \in \mathbb{R}^d$ and all $i \in [N], j \in [M]$,*
$$\|\nabla f_{i,j}(x_1) - \nabla f_{i,j}(x_2)\| \leq L\|x_1 - x_2\|.$$

Similar to the proof with $K = 1$, we first prove 2 lemmas related to the function decent of each step. The following lemma is very similar to Lemma 7 except the last term, since in the general case, we do not have Lemma 4.

**Lemma 9.** *For any $r \geq 0$ and any $\lambda > 0$, we have*
$$0 \leq -\eta_g \mathbb{E}_r[f(x^{r+1}) - f(x^*)] + \frac{\eta_g}{2L\lambda}\mathbb{E}_r\|g^r - \nabla f(x^r)\|^2 - \frac{1}{2}\mathbb{E}_r\|x^{r+1} - x^*\|^2 + \frac{1}{2}\|x^r - x^*\|^2$$
$$\left(\frac{\eta_g L(\lambda+1)}{2} - \frac{1}{2}\right)\mathbb{E}_r\|x^{r+1} - x^r\|^2 + \eta_g\langle\nabla f(x^r) - g^r, x^r - x^*\rangle.$$
*Here, we define $g^{-1} = \nabla f(x^{-1}) = 0$.*

*Proof.* For any $r \geq 1$, we have
$$\eta_g(f(x^r) - f(x^*))$$
$$\leq \eta_g\langle\nabla f(x^r), x^r - x^*\rangle$$
$$= \eta_g\langle\nabla f(x^r) - (\nabla f(x^r) - g^r), x^r - x^*\rangle + \eta_g\langle\nabla f(x^r) - g^r, x^r - x^*\rangle$$
$$= \eta_g\mathbb{E}_r\langle g^r, x^r - x^*\rangle + \eta_g\langle\nabla f(x^r) - g^r, x^r - x^*\rangle$$
$$= \eta_g\mathbb{E}_r\langle g^r, x^r - x^{r+1}\rangle + \eta_g\mathbb{E}_r\langle g^r, x^{r+1} - x^*\rangle + \eta_g\langle\nabla f(x^r) - g^r, x^r - x^*\rangle$$
$$\leq \eta_g\mathbb{E}_r\langle g^r, x^r - x^{r+1}\rangle - \frac{1}{2}\mathbb{E}_r\|x^{r+1} - x^r\|^2 + \frac{1}{2}\mathbb{E}_r\|x^r - x^*\|^2 - \frac{1}{2}\mathbb{E}_r\|x^{r+1} - x^*\|^2$$
$$+ \eta_g\langle\nabla f(x^r) - g^r, x^r - x^*\rangle.$$
When $r = 0$, the inequality also holds. We also have
$$\eta_g\mathbb{E}_r\langle g^r, x^r - x^{r+1}\rangle$$
$$= \eta_g\mathbb{E}_r\langle g^r - \nabla f(x^r), x^r - x^{r+1}\rangle + \eta_g\mathbb{E}_r\langle\nabla f(x^r), x^r - x^{r+1}\rangle$$
$$\leq \frac{\eta_g}{2\lambda L}\mathbb{E}_r\|g^r - \nabla f(x^r)\|^2 + \frac{\eta_g\lambda L}{2}\mathbb{E}_r\|x^r - x^{r+1}\|^2 + \eta_g(f(x^r) - f(x^{r+1})) + \frac{\eta_g L}{2}\|x^{r+1} - x^r\|^2.$$
Summing up the two inequalities we conclude the proof. $\qquad\square$

**Lemma 10.** *For any $r \geq 0$ and any $\lambda > 0$, we have*
$$0 \leq \eta_g(f(x^r) - f(x^{r+1})) + \frac{\eta_g}{2L\lambda}\|g^r - f(x^r)\|^2 + \left(\frac{\eta_g L(\lambda+1)}{2} - 1\right)\|x^r - x^{r+1}\|^2.$$

*Proof.*
$$0 = \eta_g\langle g^r, x^r - x^{r+1}\rangle + \eta_g\langle g^r, x^{r+1} - x^r\rangle$$
$$= \eta_g\langle\nabla f(x^r), x^r - x^{r+1}\rangle + \eta_g\langle g^r - \nabla f(x^r), x^r - x^{r+1}\rangle - \|x^{r+1} - x^r\|^2$$
$$\leq \eta_g(f(x^r) - f(x^{r+1})) + \eta_g\langle g^r - \nabla f(x^r), x^r - x^{r+1}\rangle + \left(\frac{\eta_g L}{2} - 1\right)\|x^{r+1} - x^r\|^2$$
$$\leq \eta_g(f(x^r) - f(x^{r+1})) + \frac{\eta_g}{2\lambda L}\|g^r - \nabla f(x^r)\|^2 + \left(\frac{\eta_g L(\lambda+1)}{2} - 1\right)\|x^{r+1} - x^r\|^2.$$
$\qquad\square$

Then we bound the inner product term.

**Lemma 11.** *For any $t \geq 2$ and any $c, c' > 0$, we have*

$$\sum_{r=1}^{t} \mathbb{E}\langle \nabla f(x^r) - g^r, x^r - x^* \rangle$$

$$\leq \frac{1}{2p} \mathbb{E} \sum_{r=1}^{t} \left( c\|\nabla f(x^{r-1}) - g^{r-1}\|_2^2 + \frac{1}{c}\|x^r - x^{r-1}\|_2^2 + \frac{c'}{KS} \sum_{k=0}^{K} \sum_{i \in S^r} \|g_{i,k}^r - g_{i,0}^r\|^2 + \frac{1}{c'}\|x^{r-1} - x^*\|^2 \right).$$

*Proof.*

$$\mathbb{E}\langle \nabla f(x^r) - g^r, x^r - x^* \rangle$$
$$=\mathbb{E}\langle \nabla f(x^r) - g^r, x^r - x^{r-1} \rangle + \mathbb{E}\langle \nabla f(x^r) - g^r, x^{r-1} - x^* \rangle$$
$$\leq \frac{c}{2}\mathbb{E}\|f(x^r) - g^r\|_2^2 + \frac{1}{2c}\mathbb{E}\|x^r - x^{r-1}\|_2^2 + \mathbb{E}\langle \nabla f(x^r) - g^r, x^{r-1} - x^* \rangle,$$

where the last inequality comes from Eq. (4). We also have

$$\mathbb{E}\langle \nabla f(x^r) - g^r, x^{r-1} - x^* \rangle$$
$$=(1-p)\mathbb{E}\langle f(x^{r-1}) - g^{r-1}, x^{r-1} - x^* \rangle + \mathbb{E}\langle \nabla f(x^r) - g^r - (1-p)(\nabla f(x^{r-1} - g^{r-1}), x^{r-1} - x^* \rangle.$$

Then we can compute the second term in the previous inequality.

$$\mathbb{E}\langle \nabla f(x^r) - g^r - (1-p)(\nabla f(x^{r-1}) - g^{r-1}), x^{r-1} - x^* \rangle$$

$$=(1-p)\mathbb{E}\left\langle \nabla f(x^r) - \frac{1}{KS}\sum_{k=0}^{K}\sum_{i \in S^r} g_{i,k}^r + \frac{1}{S}\sum_{i \in S^r} g_{i,0}^r - \frac{1}{S}\sum_{i \in S^r} g_{i,0}^r + g^{r-1} - \nabla f(x^{r-1}), x^{r-1} - x^* \right\rangle$$

$$=(1-p)\mathbb{E}\left\langle -\frac{1}{KS}\sum_{k=0}^{K}\sum_{i \in S^r} g_{i,k}^r + \frac{1}{S}\sum_{i \in S^r} g_{i,0}^r, x^{r-1} - x^* \right\rangle + \mathbb{E}\left\langle \nabla f(x^r) - \frac{1}{S}\sum_{i \in S^r} g_{i,0}^r + g^{r-1}, x^{r-1} - x^* \right\rangle$$

$$=(1-p)\mathbb{E}\left\langle -\frac{1}{KS}\sum_{k=0}^{K}\sum_{i \in S^r} g_{i,k}^r + \frac{1}{S}\sum_{i \in S^r} g_{i,0}^r, x^{r-1} - x^* \right\rangle$$

$$\leq \frac{c'}{2}\|\frac{1}{KS}\sum_{k=0}^{K}\sum_{i \in S^r} g_{i,k}^r - \frac{1}{S}\sum_{i \in S^r} g_{i,0}^r\|^2 + \frac{1}{2c'}\|x^{r-1} - x^*\|^2,$$

where we use the fact that $\mathbb{E}\left[\nabla f(x^r) - \frac{1}{S}\sum_{i \in S^r} g_{i,0}^r + g^{r-1} - \nabla f(x^{r-1})\right] = 0$ and Eq. (4).

Combining the computations together, we get for any $c, c' > 0$,

$$\mathbb{E}\langle \nabla f(x^r) - g^r, x^r - x^* \rangle$$

$$\leq(1-p)\mathbb{E}\langle f(x^{r-1}) - g^{r-1}, x^{r-1} - x^* \rangle + \frac{c}{2}\mathbb{E}\|f(x^r) - g^r\|_2^2 + \frac{1}{2c}\mathbb{E}\|x^r - x^{r-1}\|_2^2$$

$$+ \frac{c'}{2}\|\frac{1}{KS}\sum_{k=0}^{K}\sum_{i \in S^r} g_{i,k}^r - \frac{1}{S}\sum_{i \in S^r} g_{i,0}^r\|^2 + \frac{1}{2c'}\|x^{r-1} - x^*\|^2$$

$$\leq(1-p)\mathbb{E}\langle f(x^{r-1}) - g^{r-1}, x^{r-1} - x^* \rangle + \frac{c}{2}\mathbb{E}\|f(x^r) - g^r\|_2^2 + \frac{1}{2c}\mathbb{E}\|x^r - x^{r-1}\|_2^2$$

$$+ \frac{c'}{2KS}\sum_{k=0}^{K}\sum_{i \in S^r} \|g_{i,k}^r - \sum_{i \in S^r} g_{i,0}^r\|^2 + \frac{1}{2c'}\|x^{r-1} - x^*\|^2.$$

We also know that $\mathbb{E}\langle \nabla f(x^0) - g^0, x^0 - x^* \rangle = 0$, and we can get for any $t \geq 1$,

$$\sum_{r=1}^{t} \mathbb{E}\langle \nabla f(x^r) - g^r, x^r - x^* \rangle$$

$$\leq \frac{1}{2p} \mathbb{E} \sum_{r=1}^{t} \left( c \|\nabla f(x^{r-1}) - g^{r-1}\|_2^2 + \frac{1}{c} \|x^r - x^{r-1}\|_2^2 + \frac{c'}{KS} \sum_{k=0}^{K} \sum_{i \in S^r} \|g_{i,k}^r - g_{i,0}^r\|^2 + \frac{1}{c'} \|x^{r-1} - x^*\|^2 \right).$$

$\square$

**Lemma 12.** *For any $c, c' > 0$ such that $\frac{t\eta_g}{pc'} \leq 1/2$, we have*

$$\sum_{r=1}^{t} \mathbb{E} \langle \nabla f(x^r) - g^r, x^r - x^* \rangle$$

$$\leq \frac{t}{pc'} \|x^0 - x^*\|^2 + \left( \frac{c}{2p} + \frac{t\eta_g c}{p^2 c'} \right) \mathbb{E} \sum_{r=1}^{t} \|\nabla f(x^{r-1}) - g^{r-1}\|_2^2$$

$$+ \left( \frac{1}{2cp} + \frac{tc}{pc'} + \frac{2\eta_g t}{p^2 c'} \right) \mathbb{E} \sum_{r=1}^{t} \|x^r - x^{r-1}\|_2^2 + \left( \frac{c'}{2p} + \frac{t\eta_g}{p^2 c'} \right) \mathbb{E} \sum_{r=1}^{t} \frac{1}{KS} \sum_{k=0}^{K} \sum_{i \in S^r} \|g_{i,k}^r - g_{i,0}^r\|^2.$$

*Proof.*

$$\|x^r - x^*\|^2$$
$$= \|x^{r-1} - x^* - \eta_g g^{r-1}\|^2$$
$$= \|x^{r-1} - x^*\|^2 + \eta_g^2 \|g^{r-1}\|^2 - 2\eta_g \langle g^{r-1}, x^{r-1} - x^* \rangle$$
$$= \|x^{r-1} - x^*\|^2 + \|x^r - x^{r-1}\|^2 - 2\eta_g \langle \nabla f(x^{r-1}), x^{r-1} - x^* \rangle - 2\eta_g \langle g^{r-1} - \nabla f(x^{r-1}), x^{r-1} - x^* \rangle$$
$$\leq \|x^{r-1} - x^*\|^2 + \|x^r - x^{r-1}\|^2 + 2\eta_g (f(x^*) - f(x^{r-1})) + 2\eta_g \langle \nabla f(x^{r-1}) - g^{r-1}, x^{r-1} - x^* \rangle$$
$$\leq \|x^{r-1} - x^*\|^2 + \|x^r - x^{r-1}\|^2 + 2\eta_g \langle \nabla f(x^{r-1}) - g^{r-1}, x^{r-1} - x^* \rangle$$
$$\leq \|x^0 - x^*\|^2 + \sum_{r'=1}^{r} \|x^{r'} - x^{r'-1}\|^2 + 2\eta_g \sum_{r'=1}^{r} \langle \nabla f(x^{r'-1}) - g^{r'-1}, x^{r'-1} - x^* \rangle.$$

For simplicity, we define the following notations,

$$A_c^t = \mathbb{E} \sum_{r=1}^{t} c \|\nabla f(x^{r-1}) - g^{r-1}\|_2^2$$

$$B_c^t = \mathbb{E} \sum_{r=1}^{t} \frac{1}{c} \|x^r - x^{r-1}\|_2^2$$

$$C_{c'}^t = \mathbb{E} \sum_{r=1}^{t} \frac{c'}{KS} \sum_{k=0}^{K} \sum_{i \in S^r} \|g_{i,k}^r - g_{i,0}^r\|^2$$

$$D^t = \mathbb{E} \sum_{r=1}^{t} \|x^{r-1} - x^*\|^2$$

$$D_{c'}^t = \mathbb{E} \sum_{r=1}^{t} \frac{1}{c'} \|x^{r-1} - x^*\|^2.$$

From Lemma 11, we know that for any $t' \leq t$, we have

$$\sum_{r=1}^{t'} \mathbb{E} \langle \nabla f(x^r) - g^r, x^r - x^* \rangle \leq \frac{1}{2p} \left( A_c^{t'} + B_c^{t'} + C_{c'}^{t'} + D_{c'}^{t'} \right)$$

$$\leq \frac{1}{2p} \left( A_c^t + B_c^t + C_{c'}^t + D_{c'}^t \right),$$

and for any $r \leq t$, we can bound $\|x^r - x^*\|^2$ by

$$\mathbb{E} \|x^r - x^*\|^2$$

$$\leq \mathbb{E}\|x^0 - x^*\|^2 + \mathbb{E}\sum_{r'=1}^{r}\|x^{r'} - x^{r'-1}\|^2 + 2\eta_g\mathbb{E}\sum_{r'=1}^{r}\langle\nabla f(x^{r'-1}) - g^{r'-1}, x^{r'-1} - x^*\rangle$$

$$\leq \mathbb{E}\|x^0 - x^*\|^2 + \mathbb{E}\sum_{r'=1}^{r}\|x^{r'} - x^{r'-1}\|^2 + 2\eta_g\frac{1}{2p}\left(A_c^r + B_c^r + C_{c'}^r + D_{c'}^r\right)$$

$$\leq \mathbb{E}\|x^0 - x^*\|^2 + cB_c^t + \frac{\eta_g}{p}\left(A_c^t + B_c^t + C_{c'}^t + D_{c'}^t\right).$$

Then we bound $D^t$, we have

$$D^t = \sum_{r=1}^{t}\|x^{r-1} - x^*\|^2$$

$$\leq \sum_{r=1}^{t}\left(\mathbb{E}\|x^0 - x^*\|^2 + cB_c^t + \frac{\eta_g}{p}\left(A_c^t + B_c^t + C_{c'}^t + D_{c'}^t\right)\right)$$

$$\leq t\|x^0 - x^*\|^2 + \frac{t\eta_g}{p}A_c^t + t\left(c + \frac{\eta_g}{p}\right)B_c^t + \frac{t\eta_g}{p}C_{c'}^t + \frac{t\eta_g}{pc'}D^t.$$

As long as $\frac{t\eta_g}{pc'} \leq 1/2$, we have

$$D^t \leq 2t\|x^0 - x^*\|^2 + \frac{2t\eta_g}{p}A_c^t + 2t\left(c + \frac{\eta_g}{p}\right)B_c^t + \frac{2t\eta_g}{p}C_{c'}^t.$$

Then we plug this inequality into Lemma 11, we get

$$\sum_{r=1}^{t}\mathbb{E}\langle\nabla f(x^r) - g^r, x^r - x^*\rangle$$

$$\leq \frac{1}{2p}\mathbb{E}\sum_{r=1}^{t}\left(c\|\nabla f(x^{r-1}) - g^{r-1}\|_2^2 + \frac{1}{c}\|x^r - x^{r-1}\|_2^2 + \frac{c'}{KS}\sum_{k=0}^{K}\sum_{i\in S^r}\|g_{i,k}^r - g_{i,0}^r\|^2 + \frac{1}{c'}\|x^{r-1} - x^*\|^2\right).$$

$$\leq \frac{1}{2p}\mathbb{E}\sum_{r=1}^{t}\left(c\|\nabla f(x^{r-1}) - g^{r-1}\|_2^2 + \frac{1}{c}\|x^r - x^{r-1}\|_2^2 + \frac{c'}{KS}\sum_{k=0}^{K}\sum_{i\in S^r}\|g_{i,k}^r - g_{i,0}^r\|^2\right)$$

$$+ \frac{1}{2pc'}\left(2t\|x^0 - x^*\|^2 + \frac{2t\eta_g}{p}A_c^t + 2t\left(c + \frac{\eta_g}{p}\right)B_c^t + \frac{2t\eta_g}{p}C_{c'}^t\right)$$

$$\leq \frac{t}{pc'}\|x^0 - x^*\|^2 + \left(\frac{c}{2p} + \frac{t\eta_g c}{p^2 c'}\right)\mathbb{E}\sum_{r=1}^{t}\|\nabla f(x^{r-1}) - g^{r-1}\|_2^2$$

$$+ \left(\frac{1}{2cp} + \frac{t}{pc'} + \frac{\eta_g t}{p^2 cc'}\right)\mathbb{E}\sum_{r=1}^{t}\|x^r - x^{r-1}\|_2^2 + \left(\frac{c'}{2p} + \frac{t\eta_g}{p^2}\right)\mathbb{E}\sum_{r=1}^{t}\frac{1}{KS}\sum_{k=0}^{K}\sum_{i\in S^r}\|g_{i,k}^r - g_{i,0}^r\|^2.$$

$$\square$$

**Theorem 2** (Convergence of FedPAGE in convex setting). *Under Assumption 1 (and Assumption 2), if we choose the sampling probability $p_r = p = \frac{S}{N}$ for every $r \geq 1$ and $p_0 = 1$, the minibatch sizes $b_1 = \min\{M, \frac{24\sigma^2}{p^{1/2}\sqrt{S}\epsilon}\}, b_2 = \min\{M, \frac{48\sigma^2}{p^{3/2}\sqrt{S}\epsilon}\}$, and the global and local step size*

$$\eta_g = \Theta\left(\frac{(S + N^{3/4})\frac{S}{N}}{L(S + \sqrt{N})}\right), \qquad \eta_l = O\left(\frac{S}{N^{5/4}KL_c\sqrt{T}}\right),$$

*then* FedPAGE *satisfies*

$$\frac{1}{R}\sum_{r=0}^{R-1}\mathbb{E}[f(x^{r+1}) - f(x^*)] \leq \begin{cases} O\left(\frac{N^{3/4}L}{SR} + \epsilon\right), \text{if } S \leq \sqrt{N} \\ O\left(\frac{N^{1/4}L}{R} + \epsilon\right), \text{if } \sqrt{N} < S \leq N^{3/4} \\ O\left(\frac{NL}{SR} + \epsilon\right), \text{if } N^{3/4} < S \end{cases}.$$

*Proof.* First we have

$$\eta_g \mathbb{E}_r[f(x^{r+1}) - f(x^*) + \delta(f(x^{r+1}) - f(x^r))]$$

$$\leq \frac{\eta_g(1+\delta)}{2L\lambda} \mathbb{E}_r \|g^r - \nabla f(x^r)\|^2 - \frac{1}{2} \mathbb{E}_r \|x^{r+1} - x^*\|^2 + \frac{1}{2} \|x^r - x^*\|^2$$

$$\left( \frac{\eta_g L(\lambda+1)(1+\delta)}{2} - \frac{1+2\delta}{2} \right) \mathbb{E}_r \|x^{r+1} - x^r\|^2 + \eta_g \langle \nabla f(x^r) - g^r, x^r - x^* \rangle.$$

Summing up the inequality and choosing $b_1 = \min\{M, \frac{24\sigma^2\sqrt{S}}{\epsilon p^{3/2} N}\}, b_2 = \min\{M, \frac{48\sigma^2\sqrt{S}}{\epsilon p^{3/2} S}\}$,

$$\sum_{r=0}^{T-1} \eta_g \mathbb{E}[f(x^{r+1}) - f(x^*)] + \eta_g \delta \mathbb{E}[f(x^T) - f(x^0)]$$

$$\leq -\frac{1}{2} \mathbb{E}\|x^T - x^*\|^2 + \frac{1}{2}\|x^0 - x^*\|^2 + \sum_{r=0}^{T-1} \frac{\eta_g(1+\delta)}{2L\lambda} \mathbb{E}\|g^r - \nabla f(x^r)\|^2$$

$$+ \sum_{r=0}^{T-1} (1+\delta) \frac{L(\lambda+1)\eta_g - 1}{2} \mathbb{E}\|x^{r+1} - x^r\|^2 + \sum_{r=0}^{T-1} \eta_g \mathbb{E}\langle \nabla f(x^{r-1}) - g^{r-1}, x^r - x^* \rangle$$

$$\leq -\frac{1}{2} \mathbb{E}\|x^T - x^*\|^2 + \frac{1}{2}\|x^0 - x^*\|^2 + \sum_{r=0}^{T-1} \left( \frac{\eta_g(1+\delta)}{2L\lambda} + \frac{c\eta_g}{2p} + \frac{T\eta_g^2 c}{p^2 c'} \right) \mathbb{E}\|g^r - \nabla f(x^r)\|^2$$

$$+ \sum_{r=0}^{T-1} (1+\delta) \left( \frac{L\eta_g(\lambda+1) - 1}{2} + \frac{\eta_g}{2cp(1+\delta)} + \frac{T\eta_g}{pc'(1+\delta)} + \frac{\eta_g^2 T}{p^2 cc'(1+\delta)} \right) \mathbb{E}\|x^{r+1} - x^r\|^2$$

$$+ \left( \frac{c'\eta_g}{2p} + \frac{T\eta_g^2}{p^2} \right) \mathbb{E} \sum_{r=0}^{T-1} \frac{1}{KS} \sum_{k=0}^{K} \sum_{i\in S^r} \|g_{i,k}^r - g_{i,0}^r\|^2 + \frac{T\eta_g}{pc'}\|x^0 - x^*\|^2$$

$$\leq -\frac{1}{2} \mathbb{E}\|x^T - x^*\|^2 + \frac{1}{2}\|x^0 - x^*\|^2 + \frac{T\eta_g}{pc'}\|x^0 - x^*\|^2 + 6K^2 L^2 \eta_l^2 \left( \frac{c'\eta_g}{2p} + \frac{T\eta_g^2}{p^2} \right) \frac{T\sigma^2 \mathbb{I}\{b_2 < M\}}{b_2}$$

$$+ \sum_{r=0}^{T-1} w_1 \mathbb{E}\|g^r - \nabla f(x^r)\|^2$$

$$+ \sum_{r=0}^{T-1} (1+\delta) \cdot w_2 \cdot \mathbb{E}\|x^{r+1} - x^r\|^2$$

$$+ 12K^2 L^2 \eta_l^2 \left( \frac{c'\eta_g}{2p} + \frac{T\eta_g^2}{p^2} \right) \sum_{r=0}^{T-1} \|\nabla f(x^r)\|^2,$$

$$\overset{(19)}{\leq} -\frac{1}{2} \mathbb{E}\|x^T - x^*\|^2 + \frac{1}{2}\|x^0 - x^*\|^2 + \frac{T\eta_g}{pc'}\|x^0 - x^*\|^2 + 6K^2 L^2 \eta_l^2 \left( \frac{c'\eta_g}{2p} + \frac{T\eta_g^2}{p^2} \right) (\sqrt{S}p^{3/2} T\epsilon)$$

$$+ \sum_{r=0}^{T-1} \left( (1+\delta) \cdot w_2 + \frac{3}{p} \frac{1-p/3}{S} w_1 \right) \cdot \mathbb{E}\|x^{r+1} - x^r\|^2$$

$$+ \left( 12K^2 L^2 \eta_l^2 \left( \frac{c'\eta_g}{2p} + \frac{T\eta_g^2}{p^2} \right) + \frac{144K^2 L^2 \eta_l^2}{p^2} w_1 \right) \sum_{r=0}^{T-1} \|\nabla f(x^r)\|^2 + \frac{3\epsilon T p^{3/2}}{8\sqrt{S}} w_1.$$

where we define

$$w_1 := \left( \frac{\eta_g(1+\delta)}{2L\lambda} + \frac{c\eta_g}{2p} + \frac{T\eta_g^2 c}{p^2 c'} + 12K^2 L^2 \eta_l^2 \left( \frac{c'\eta_g}{2p} + \frac{T\eta_g^2}{p^2} \right) \right),$$

$$w_2 := \left( \frac{L\eta_g(\lambda+1) - 1}{2} + \frac{\eta_g}{2cp(1+\delta)} + \frac{T\eta_g}{pc'(1+\delta)} + \frac{\eta_g^2 T}{p^2 cc'(1+\delta)} + \frac{12K^2 L^2 \eta_l^2}{1+\delta} \left( \frac{c'\eta_g}{2p} + \frac{T\eta_g^2}{p^2} \right) \right).$$

By choosing $p = \frac{S}{N}, \lambda = \sqrt{1/Sp}, c = \sqrt{Sp/L^2}, \delta = 1/(S^{1/4}p^{3/4}), \eta_g = O\left(\frac{(1+\delta)p}{L(1+\sqrt{1/(Sp)})}\right), c' = \frac{2T\eta_g}{p}, \eta_l = O\left(\frac{S}{N^{5/4}KL\sqrt{T}}\right)$, we get

$$
\begin{aligned}
\frac{\eta_g(1+\delta)}{2L\lambda} &= (1+\delta)O\left(\frac{(1+\delta)S/N\cdot S}{L^2(1+\sqrt{N}/S)\sqrt{N}}\right) & = (1+\delta)O\left(\frac{S^2}{N^{4/5}L^2}\right), \\
\frac{c\eta_g}{p} &= (1+\delta)O\left(\frac{S}{L^2\sqrt{N}(1+\sqrt{N}/S)}\right) & = (1+\delta)O\left(\frac{S^2}{NL^2}\right), \\
\frac{T\eta_g^2 c}{p^2 c'} &= (1+\delta)O\left(\frac{S}{L^2 N(1+\sqrt{N}/S)}\right) & = (1+\delta)O\left(\frac{S^2}{N^{3/2}L^2}\right), \\
\frac{c'\eta_g}{2p} + \frac{T\eta_g^2}{p^2} &= 2\frac{T\eta_g^2}{p^2} & = O\left(\frac{2T\sqrt{N}}{L^2}\right).
\end{aligned}
$$

Then, we can verify that $w_1 \cdot N/S^2 = (1+\delta)\cdot O(1)$. Similar to the proof of Theorem 4, we can also verify that $w_2 = O(1)$, and we can choose $\eta_g$ and $\eta_l$ with a small constant such that

$$
\left((1+\delta)\cdot w_2 + \frac{3}{p}\frac{1-p/3}{S}w_1\right) \leq 0,
$$

$$
6K^2L^2\eta_l^2\left(\frac{c'\eta_g}{2p} + \frac{T\eta_g^2}{p^2}\right)(\sqrt{S}p^{3/2}T\epsilon) \leq \frac{\eta_g\epsilon T}{16},
$$

$$
\left(12K^2L^2\eta_l^2\left(\frac{c'\eta_g}{2p} + \frac{T\eta_g^2}{p^2}\right) + \frac{144K^2L^2\eta_l^2}{p^2}w_1\right) \leq \frac{\eta_g}{4L},
$$

$$
\frac{3\epsilon Tp^{3/2}}{8\sqrt{S}}w_1 \leq \frac{3\eta_g\epsilon T}{8}.
$$

Then we have

$$
\sum_{r=0}^{T-1}\mathbb{E}[f(x^{r+1}) - f(x^*)]
$$

$$
\leq \delta\mathbb{E}[f(x^0) - f(x^T)] + \frac{1}{\eta_g}\|x^0 - x^*\|^2 + \frac{\epsilon T}{16} + \frac{1}{4L}\sum_{r=0}^{T-1}\|\nabla f(x^r)\|^2 + \frac{3\epsilon T}{8}
$$

$$
\leq \delta\mathbb{E}[f(x^0) - f(x^T)] + \frac{1}{\eta_g}\|x^0 - x^*\|^2 + \frac{1}{2}\sum_{r=0}^{T-1}\mathbb{E}[f(x^r) - f(x^*)] + \frac{7\epsilon T}{16}.
$$

Then we know that

$$
\frac{1}{T}\sum_{r=0}^{T-1}\mathbb{E}[f(x^{r+1}) - f(x^*)] \leq 2\frac{\delta}{T}(f(x^0) - f(x^*)) + \frac{2}{\eta_g T}\|x^0 - x^*\|^2 + \frac{7\epsilon}{16}.
$$

Recall that

$$
\eta_g = \Theta\left(\frac{(1+\frac{N^{3/4}}{S})\frac{S}{N}}{L(1+\frac{\sqrt{N}}{S})}\right) = \Theta\left(\frac{(S+N^{3/4})\frac{S}{N}}{L(S+\sqrt{N})}\right) = \begin{cases} \Theta\left(\frac{S}{N^{3/4}L}\right), & \text{if } S \leq \sqrt{N} \\ \Theta\left(\frac{1}{N^{1/4}L}\right), & \text{if } \sqrt{N} < S \leq N^{3/4} \\ \Theta\left(\frac{S}{NL}\right), & \text{if } N^{3/4} < S \end{cases}.
$$

We have

$$\frac{1}{R}\sum_{r=0}^{R-1}\mathbb{E}[f(x^{r+1}) - f(x^*)] \leq \begin{cases} O\left(\dfrac{N^{3/4}L}{SR} + \epsilon\right), \text{ if } S \leq \sqrt{N} \\ O\left(\dfrac{N^{1/4}L}{R} + \epsilon\right), \text{ if } \sqrt{N} < S \leq N^{3/4} \\ O\left(\dfrac{NL}{SR} + \epsilon\right), \text{ if } N^{3/4} < S \end{cases}.$$

$\square$

