# OpenReview forum: "FedPAGE: A Fast Local Stochastic Gradient Method for Communication-Efficient Federated Learning"
_ICLR.cc/2022/Conference — ICLR 2022 Submitted_

### Official Review · Reviewer_uLBz · 2021-11-01

**Correctness:** 4
**Technical Novelty And Significance:** 3
**Empirical Novelty And Significance:** 2
**Recommendation:** 5
**Confidence:** 4

**Main Review:**

- The theoretical part of the paper is relatively well-written and clear.
- The overall contribution of the paper is very clear. The paper takes a recent distributed algorithm with a good convergence rate, and adds local computation to the algorithm (which is very common in FL), and shows it still achieves a good convergence rate. And because the original algorithm has superior convergence results, the FedPAGE also enjoys the same superiority over other analyzed FL methods. But if one steps back a bit here, actually the convergence guarantee of the FedPAGE algorithm is not better than the PAGE algorithm. This is clear from the approach in the proof (Lemma 1) and Table 3, where K=1 would give the best convergence guarantee for FedPAGE. This is important because adding local updates is at the heart of most FL algorithms despite the fact that there is little theoretical evidence that they improve the method. Unfortunately, these points are not very clearly conveyed in the main text. Also, due to this fact, it is important to accompany the theoretical results of the paper with solid experimental results that can support the use of local steps in FedPAGE under different circumstances (heterogeneity levels) and against many solid FL algorithms. As I will point out later, this part is very much lacking in the paper.
- It seems that the paper assumes the same number of data points per client and I am not sure if this can be easily relaxed. Such an assumption is unrealistic because in FL clients are heterogeneous.
- One part of the FedPAGE that in my opinion makes it basically impractical (for large N practical FL problems) is the need for rounds that communicate with all the nodes. In spite of the fact that these rounds are very rare (due to low p) during the run of the algorithm, they are not possible in practice mainly because 1) not all devices are available at the same time for an update (update usually happen only when devices are on wifi and connected to power).  And 2) the delay for the responses has a very long tail; meaning that waiting for all the devices to respond in a round takes a long time. And 3) the devices can drop out during any round which means that you cannot count on all devices for a round. This means that the few rounds that need complete device participation might never finish (or take a very long time to do so).
- It is not very clear to me why in FedPAGE the effective step-size is $\eta_g$ (as opposed to FedAvg and SCAFFOLD). It would be great if the authors can elaborate.
- Regarding the experiments (For the importance of the experimental section for this paper see my first point):
  - No comparison is done at different heterogeneity levels of data (which are known to have a great effect on FL algorithms).
  - Not a single result is reported for the achieved test accuracy or other metrics that are relevant to ML
  - For the comparison against other FL methods, no comparison is done for state-of-the-art methods such as the one that uses momentum, ... ! SCAFFOLD and FedAvg are not empirically the state-of-the-art methods.
  - I had a very hard time understanding how the HP tuning has happened. What parameters were tuned for each algorithm and how (what were the metrics and possible values). This is important in all the results but esp important for Fig 2 and 3 where there are other algorithms and their HPs are also involved and the comparison needs to be fair.

It is worth noting that I did not check the proofs of the paper thoroughly, but the claims and results seem reasonable and correct.

Minor comments:

- last paragraph of the introduction is not very well connected to the rest of the intro.
- The algorithm does not summarize how a solution (x) is returned. I had to go through the appendix theorems and proofs to understand that the authors assume the return of one of the x^r (probably at random based on the convergence guarantee).

===== After rebuttal =====

I read the responses and other reviews/comments and have decided to reduce my score to 5. I agree with other reviewers that the paper has flaws and issues that have not been addressed and it is not ready to be published. I encourage the authors to strengthen the theoretical results by looking at the possible benefits of local steps and also include more informative experiments.



**Summary Of The Paper:**

The paper considers a federated variant of the PAGE algorithm, called FedPAGE that utilizes local steps. The authors analyze the convergence behavior of FedPAGE algorithm and show that it is very similar to PAGE. And as PAGE has good convergence guarantees, compared to other distributed optimization methods, FedPAGE shows good convergence results compared to other federated learning methods with guarantees. The authors have a limited set of experiments to compare FedPAGE with other federated methods and also PAGE.

**Summary Of The Review:**

Based on the above points, I would say the current version of the paper is slightly above the acceptance threshold. I would be willing to increase the paper's score based on the authors' responses/updates to my comments/concerns (mainly around novelty, assumptions, practicality, and experiments).

---

### Official Review · Reviewer_Xsi6 · 2021-11-01

**Correctness:** 3
**Technical Novelty And Significance:** 2
**Empirical Novelty And Significance:** 2
**Recommendation:** 3
**Confidence:** 4

**Main Review:**

1. Table 1: the convergence round does not account for the full-participation round. a) This communication round should be included as communication rounds b) the comparison with other algorithms is *unfair* since the other algorithms never allow for a full participation round.
    - To make the above point clearer, let us consider Distributed-SVRG (with, say only 1 local step). Distributed-SVRG only requires O(1/ε) rounds of full participation, and O(L/ε) rounds of sampling. The total communication round is only O(L/ε).

2. The paper assumes a strong, non-standard assumption (in FL study) that each client can access the *deterministic* gradient oracle of the local objective. The table 1 compares their results with FedAvg / SCAFFOLD, but all these methods are tailored for stochastic gradient oracle.

3. Significancy: The main Theorem 1 and Theorem 2, are trivial applications of the PAGE with no local steps. The argument follows by the statement that running K > 1 of local steps are not worse than running K = 1 step. As a result of this trivial application, running K>1 local step gives no improvement.

===

I read the response and decided to keep my initial score.

**Summary Of The Paper:**

This paper proposes FedPAGE, a federated variant of PAGE, which is claimed to attain the SOTA communication efficiency theoretically.

**Summary Of The Review:**

 My main concern of this work lies in the significancy of the results, and improper comparison with other algorithms.

---

### Official Review · Reviewer_tXTB · 2021-11-07

**Correctness:** 4
**Technical Novelty And Significance:** 3
**Empirical Novelty And Significance:** 3
**Recommendation:** 5
**Confidence:** 3

**Main Review:**

Strengths:
1)  Adapting PAGE algorithm into federated setting with probabilistic local updates reduces the communication complexity on both federated convex and nonconvex optimization.
2) Numerical experiments demonstrates the effectiveness of multiples local update steps.

Weaknesses:
1) Lack of analysis on test performance. Numerical experiments show that FedPAGE converges much faster than the comparison methods. How is the performance on test settings? Does the proposed optimization method achieve comparable or even better performance beyond convergence speed?

**Summary Of The Paper:**

This paper proposes a new federated learning algorithm to reduce the communication cost on both federated convex and nonconvex optimization by applying optimal PAGE method into the federated setting and running multiple local update steps on clients before communicating to the orchestrating server. Experiments verifies the effectiveness of multiple local update steps and its superiority over other methods.
Contributions:
1)  Adapting PAGE algorithm into federated setting with probabilistic local updates reduces the communication complexity on both federated convex and nonconvex optimization.
2) Numerical experiments demonstrates the effectiveness of multiples local update steps.


**Summary Of The Review:**

 Reducing the communication cost  with multiple local updates by applying PAGE in federated optimization.

---

### Official Review · Reviewer_oGr1 · 2021-11-08

**Correctness:** 4
**Technical Novelty And Significance:** 3
**Empirical Novelty And Significance:** 3
**Recommendation:** 5
**Confidence:** 3

**Main Review:**

### Strengths
1. FedPAGE can handle larger effective learning rate.
2. Theoretiacally FedPAGE improves over variance reduction-based SCAFFOLD algorithm in both smooth convex and smooth non-convex settings.
3. Empirically FedPAGE is better than FedAvg. It is also comparable or better than SCAFFOLD.

### Weaknesses
1. Table 1: What should we focus on total communication bandwidth (S*R) or number of communication rounds (R)?

2. Table 1:  Assumption $S \leq \sqrt{N}$ should be explicitely stated. Otherwise, FedPAGE may even only match SCAFFOLD’s round complexity.

3. Not a lot of discussion on the limitations.

i. SCAFFOLD never samples all clients at the synchronization rounds. That could be a too expensive waiting for this many clients to communication. Is this a necessity for FedPAGE? Is it possible to weaken this requirement?

ii. b1 and b2 are large. Negative effects of these are not discussed in this text. For example in convex, since b3=1, effective K=K+b2=K+ min(M,N^3/2/S^2). Even for S=\sqrt{N}, this is equal to K+min(M,\sqrt{N}). It is expected that authors discuss all such  cases.

4. Experimental results are not comprehensive enough.

i. even though non-convex rate seems to be the main highlighted theoretical results, experiments are for convex optimization.

ii. It is not clear if comparing the algorithms at the same effective LR is appropriate, ideally they should all be tuned separately and the best stepsize choices for each algorithm should be compared against each other.

iii. Can the authors please provide comparison with more standard benchmark datasets like EMNIST.

iv. Since, in theory, FedPAGE is not improving PAGE, more comprehensive experiments could be useful for making a case for the former.

5. Assumption b3=1 should be explicitly stated in the theorems.

6. There are no rate comparison to non-local method like Accelerated SGD, PAGE, etc. From the results, FedPAGE and PAGE seems to have the exactly same round complexity. So it is not clear if the given theory is non-trivial. But the limited experiments seem promising since they show a constant but diminishing improvement when increasing K.

7. It would useful if the authors can provide a discussion of any technical novelty in their analysis.

### Other
1. Table 1: What is this expectation $\mathbb{E}[f]$ over?

————————————————————————
### FINAL:

I thank the authors for the response. However, after reading all the reviews, author response and public comments carefully, I have concluded that in this current state this paper cannot be publishable at this venue. Overall it seems there are many holes in the arguments of the paper. These are some additional comments detailing my reasoning.

1. I acknowledge that the authors want to position this as a theoretical paper. However, in my opinion most of the theory of FL algorithms fail to explain their success in practice. Because of this it is hard to accept the usefulness of the algorithms or the theory without extensive experimental results. A prime example for theory not explaining practice is present in this paper itself! PAGE (FedPAGE with K=1) has the same convergence rate as FedPAGE (K != 1). This is possibly due to the fact that the proofs mainly show that FedPAGE is not worse than PAGE. However, experiments show that FedPAGE (K != 1) may be faster than PAGE. In fact this has been pointed out by other reviewers too. This is never addressed by the authors.

2. Most of my concerns about the experiments are not addressed? Authors promise new experiments but doesn’t provide plots or data.

3. I can agree with the fact that large(could be full)-batch local computations may be okay when optimizing number of communication rounds. But ideally this limitation should be discussed upfront in the paper. Public commenter “Durmus Alp Emre Acar” also raises an important point where SACFFOLD (with full-gradient and K=1) gets better rate than what is given in Table 1 for convex problems! Therefore the experimental and theoretical comparisons may not be fair.

4.  Critical concern of Reviewer Gwh9 about the proof is not addressed.

5. New concern: It is not clear were “Assumption 2” is being used in the proofs.

6. Thanks for pointing out the technical novelty, although they are a bit limited.


**Summary Of The Paper:**

This paper provides a new federated optimization method: FedPAGE, based on the recent variance-reduction based PAGE method. Latter is designed to solves smooth non-convex optimization problem. FedPAGE improves over variance reduction-based SCAFFOLD algorithm in both smooth convex and smooth non-convex settings. Finally the paper provides some numerical experiments to compare FedPAGE, PAGE, FedAvg, and SCAFFOLD.

**Summary Of The Review:**

Results seems promising. However there isn’t enough discussion of limitations, novelty, and comparison to some baselines.

---

### Official Review · Reviewer_Gwh9 · 2021-11-14

**Correctness:** 3
**Technical Novelty And Significance:** 2
**Empirical Novelty And Significance:** 2
**Recommendation:** 5
**Confidence:** 4

**Main Review:**

The paper contains following strengths and weaknesses.

Strength:
- The use of PAGE estimator in federated learning is new. However, this idea is rather incremental as it is a direct application of the PAGE estimator to local method in federated learning.
- The convergence results for nonconvex setting is better than existing results in terms of dependence on the number of workers.

Weaknesses:
- The result in convex setting is not rigorous enough. See details below.
- The method requires to have communication to all clients for every $\frac{1}{p_r}$ iterations on average. This is similar to the full participation requirements for methods like FedSplit or FedPD. This appears to be not realistic as clients in federated learning might get disconnected from the server. It is better to just use a subset of clients to update as in FedAvg.
- Numerical experiment only consider one nonconvex objective while theoretical results include both convex and nonconvex. Also, only SCAFFOLD and FedAvg.

Regarding the theoretical results for convex setting, in the proof of Theorem 4, page 28, first the authors need $\eta_g \leq O\left( \frac{(1+\delta)p}{L(1+\sqrt{1/(Sp)}} \right)$ and select some small constant to satisfy the condition that $w$ is non-positive. However, when deriving the rate for communication round, they actually use $\eta_g \leq \Theta\left( \frac{(1+\delta)p}{L(1+\sqrt{1/(Sp)}} \right)$, i.e. $\eta_g$ has the same order of the term in the bracket, and plug in the choice of $\delta$ and $p$. How can you guarantee that when $\eta_g$ is in the same order of $\frac{(1+\delta)p}{L(1+\sqrt{1/(Sp)}} $, $w$ is still non-positive.

Some minor comments:
- In equation (1), it should be $\frac{1}{M}\sum_{j=1}^M$.
- Appendix, end of page 23, the first equality should be $\frac{\sigma^2}{Nb_1} \mathbb{I}[b_1 < M]$.
- Appendix, page 28, there are 3 cases for $\eta_g$ under different range of $S$, instead of $\Theta(\cdot)$, they should be $\Omega(\cdot)$ as it should be least in the order of $(\cdot)$ or higher.

**Summary Of The Paper:**

The paper proposes a new federated learning method, called FedPAGE, which uses the PAGE gradient estimator in local update and provides convergence analysis for both convex and nonconvex setting. The paper shows that the convergence rates achieved in both cases are better than existing results. Numerical experiments are included to illustrate the performance of proposed method.

**Summary Of The Review:**

The paper does have contribution towards the development of FedPAGE and its analysis for convex and nonconvex setting. While the nonconvex result is shown to be better than existing ones, the convex result is not rigorous enough to me. Numerical experiments appear to not fully verify the theoretical results as they only contain nonconvex example.

---

### Public Comment · ~Durmus_Alp_Emre_Acar1 · 2021-11-13
**On the Baseline Method Choices of FedPAGE**

Here are a couple of points for the submission:

a. *Comparison to FedProx*. Fedprox is similar to other more recent methods such as FedPD, FedDyn and FedSplit in terms of local compute levels. These methods provably and empirically outperform FedProx overwhelmingly. The results should be compared against these methods as well (Table 1-2).

b. *Local update complexity*. Authors position the paper in terms of local updates, as a way to differentiate between algorithms performing device optimization. FedPAGE has two phases in devices where (i) it initializes $g_{i,0}^r$ and $y_{i,1}^r$ with a batch size of  $b_2$ and (ii) it performs $K$ SGD steps with batch size of $b_3$. According to theorem 1&2, FedPAGE calculates exact local gradients per clients for small $\epsilon$ levels. However, the comparison is performed to the methods that do not use full gradient computation such as FedAvg, SCAFFOLD. The results should be benchmarked against other methods with similar computational overhead. For example, let us consider one pass gradient SCAFFOLD (.i.e K=1, \sigma=0). Clients compute only one full local gradients which has less local computation than FedPAGE. On the other hand, one pass gradient SCAFFOLD (‘final rate’ discussion of Lemma 15 in SCAFFOLD paper) gets a better convergence rate for convex functions (Table 1 comparison).

Thank you!

References:

FedPD: https://arxiv.org/abs/2005.11418

FedDyn: https://arxiv.org/abs/2111.04263

FedSplit: https://arxiv.org/abs/2005.05238

SCAFFOLD: https://arxiv.org/abs/1910.06378

---

### Author Response · Authors · 2021-11-23
**To all reviewers**

We thank all reviewers for the detailed and helpful comments.
Below we clarify some questions and highlight some of our contributions.

1. Regarding periodically communicating with all clients.

Although our algorithm needs to periodically communicate with all clients, the probability is very small ($p = S / N$). Here $N$ is the total number of clients, and $S$ is the sampled subset of clients to communicate within a round "normally" (the rounds do not communicate with all clients, with probability $1-p$). In expectation, it only needs to communicate with at most $2S$ clients ($S$ for normal and $S$ for the low probability event) in each round.
Moreover, in terms of the total number of communication bits, our FedPAGE algorithm is $O(N+\sqrt{N}/\epsilon^2)$ in the nonconvex setting, but the previous theoretical state-of-the-art SCAFFOLD is $O(N^{2/3}S^{1/3}/\epsilon^2)$.
**Thus our FedPAGE converges much faster than SCAFFOLD not only in terms of the communication rounds, but also in terms of the communication bits.**

Besides, there exist some standard settings in federated learning that allow people to communicate with all of the clients, e.g. cross-silo federated learning. In such settings, using a small probability to communicate with all of the clients is not an issue, and using client sampling can help to balance the communication and local computation in a more efficient way.

2. Regarding local full gradients / large batch of local stochastic gradients

First, we do not need to compute the local full gradients, and our FedPAGE algorithm supports the use of local stochastic gradients oracle. We may need to use a large batch of local gradients and the *total computation cost in terms of the variance $\sigma^2$* is possibly worse compared with SCAFFOLD. **However, the communication cost is the main bottleneck in federated learning, and local computation cost is not the main consideration here** since local methods improve the communication cost by taking many local computation steps. **Our FedPAGE algorithm achieves much better communication result and can be thought of as a trade-off between computation and communication.**


3. Technical novelty.

In addition to using the PAGE estimator on the server-side (client sampling), we also apply the recursive gradient estimator (biased estimator) on the local steps, which makes it easier to prove the convergence results.

4. Experiments.

**We think that our main contributions are theoretical results** and thus we do not use large datasets/models or observe the test performance. Moreover, **we do add more experiments in order to show that our FedPAGE algorithm can also deal with data heterogeneity**. We sort the dataset (first by the label, then by different dimensions) and run different algorithms. The performance is nearly similar in the non-shuffled case, where FedPAGE performs better than SCAFFOLD than FedAvg.


Best regards,

Authors

---

### Decision · Program_Chairs · 2022-01-20

**Decision:**

Reject

**Comment:**

This paper proposes a new federated learning method which uses the recently developed PAGE gradient estimator in the local updates, and provides convergence analysis for both convex and nonconvex loss functions. There are several technical questions raised by the reviewers that are not addressed by the author rebuttal. Given such technical issues and limited novelty and empirical evidence, I cannot recommend acceptance.